

# 1 Evaluating Simplifications of Subsurface Process
# 2 Representations for Field-scale Permafrost Hydrology Models

Bo Gao, Ethan T. Coon
Environmental Sciences Division, Oak Ridge National Laboratory, Oak Ridge, Tennessee, USA
*Correspondence to*: Ethan T. Coon (coonet@ornl.gov)
**Abstract.** Permafrost degradation within a warming climate poses a significant environmental
threat through both the permafrost carbon feedback and damage to human communities and
infrastructure. Understanding this threat relies on better understanding and numerical
representation of thermo-hydrological permafrost processes, and the subsequent accurate
prediction of permafrost dynamics. All models include simplified assumptions, implying a tradeoff
between model complexity and prediction accuracy. The main purpose of this work is to
investigate this tradeoff when applying the following commonly made assumptions: (1) assuming
equal density of ice and liquid water in frozen soil; (2) neglecting the effect of cryosuction in
unsaturated freezing soil; and (3) neglecting advective heat transport during soil freezing and thaw.
This study designed a set of 62 numerical experiments using the Advanced Terrestrial Simulator
(ATS v1.2) to evaluate the effects of these choices on permafrost hydrological outputs, including
both integrated and pointwise quantities. Simulations were conducted under different climate
conditions and soil properties from three different sites in both column- and hillslope-scale
configurations. Results showed that amongst the three physical assumptions, soil cryosuction is
the most crucial yet commonly ignored process. Neglecting cryosuction, on average, can cause 10%
~ 20% error in predicting evaporation, 50% ~ 60% error in discharge, 10% ~ 30% error in thaw
depth, and 10% ~ 30% error in soil temperature at 1 m beneath surface. The prediction error for
subsurface temperature and water saturation is more obvious at hillslope scales due to the presence
of lateral flux. By comparison, using equal ice-liquid density has a minor impact on most
hydrological variables, but significantly affects soil water saturation with an averaged 5% ~ 15%
error. Neglecting advective heat transport presents the least error, 5% or even much lower, in most
variables for a general Arctic tundra system, and can decrease the simulation time at hillslope
scales by 40% ~ 80%. By challenging these commonly made assumptions, this work provides
permafrost hydrology modelers important context for better choosing the appropriate process





representation for a given modeling experiment.
**Copyright Statement.** This manuscript has been authored by UT- Battelle, LLC under Contract
No. DE-AC05-00OR22725 with the U.S. Department of Energy. The United States Government
retains and the publisher, by accepting the article for publication, acknowledges that the United
States Government retains a non-exclusive, paid-up, irrevocable, world-wide license to publish, or
reproduce the published form of this manuscript, or allow others to do so, for United States
Government purposes. The Department of Energy will provide public access to these results of
federally    sponsored    research    in    accordance    with    the    DOE    Public    Access    Plan
(http://energy.gov/downloads/doe-public-access-plan).
**1 Introduction**
Permafrost describes a state of ground which stays frozen continuously over multiple years, which
may cover an entire region (e.g., Arctic tundra) or occur in isolation (e.g., alpine top). From the
perspective of scope, permafrost occupies approximately 23.9% (22.79 million km$^2$) of the
exposed land area of the northern hemisphere (Zhang et al., 2008), as well as alpine regions and
Antarctica in the southern hemisphere. Permafrost areas store a vast amount of organic carbon, of
which most is stored in perennially frozen soils (Hugelius et al., 2014). If the organic carbon is
exposed due to permafrost thaw, it is likely to decay with microbial activity, releasing greenhouse
gas to the atmosphere and exacerbating global warming. In Arctic tundra, permafrost also plays
an important role in maintaining water, habitat of wildlife, landscape, and infrastructure (Berteaux
et al., 2017; Dearborn et al., 2021; Hjort et al., 2018; Sugimoto et al., 2002). Permafrost
degradation may cause significant damage to the local ecosystem, reshape the surface and
subsurface hydrology, and eventually influence the global biosphere (Cheng and Wu, 2007;
Jorgenson et al., 2001; Tesi et al., 2016; Walvoord and Kurylyk, 2016). Therefore, the occurrent
and potential impacts motivate the development of computational models with the goal of better
understanding the thermal and hydrological processes in permafrost regions, and consequently
predict permafrost thaw accurately.
Simulating soil freezing and thaw processes is a challenging task that incorporates mass and energy
transfer among atmosphere, snowpack, land surface (perhaps with free water), and a variably
saturated subsurface. Several hydrological models with different complexity and applicable scales



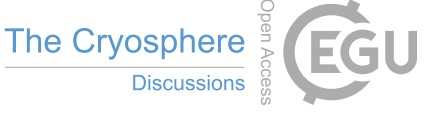

have been developed to investigate the complicated interactions. A comprehensive review of
permafrost models based on empirical and physical representations using analytical and numerical
solutions can be found in (Bui et al., 2020; Dall'Amico et al., 2011; Jan et al., 2020; Kurylyk et
al., 2014; Kurylyk and Watanabe, 2013; Riseborough et al., 2008). Process-rich models which aim
to predict permafrost change through direct simulation of mass and energy transport, such as
the Advanced Terrestrial Simulator (ATS; Painter et al., 2016), GEOtop (Endrizzi et al., 2014),
CryoGrid 3 (Westermann et al., 2016), PFLOTRAN-ICE (Karra et al., 2014), and SUTRA-ICE
(McKenzie et al., 2007), have been demonstrated to describe thermal permafrost hydrology under
various climate conditions. Nominally, representing more physical process complexity should
improve accuracy, but how much accuracy and at what computational expense (and therefore
tradeoff in ability to run larger, or larger ensembles of, simulations) is not always theoretically
known. Thus, certain assumptions or simplifications about the system are a significant part of
model development.
Even in the most process-rich models of permafrost change, three such physics simplifications are
often made: representing water at constant density (thereby neglecting the expansion of ice relative
to liquid water), neglecting cryosuction of water in unsaturated, partially frozen soils, and
neglecting advective heat transport.
First, because of the lower density of ice than liquid water, freezing water must expand the volume
of the porous media, push liquid water into nearby volume, or otherwise expand the volume
occupied by that water. As all of the current set of models operate under the assumption of a rigid
solid matrix and thus the absence of mechanical equations describing matrix deformation or frost
heave, including this expansion typically results in large pressures that must be offset by grain
compressibility or another mechanism. Therefore, the densities of ice and liquid water are
frequently assumed equal (e.g., Dall'Amico et al., 2011; Devoie and Craig, 2020; Weismüller et
al., 2011). It is uncertain whether this simplification affects predictions of permafrost change and
thermal hydrology.
Second, cryosuction describes the redistribution of water in partially frozen, unsaturated soils
caused by increased matric suction. At the interface of ice and liquid water, negative pressures
result in the migration of liquid water toward the freezing front and the subsequent increase of ice
content. Several approaches representing cryosuction in models are used (Dall'Amico et al., 2011;
Noh et al., 2012; Painter and Karra, 2014; Stuurop et al., 2021), either in an empirical form or



physically derived from the generalized Clapeyron equation. Other process-rich models have
ignored cryosuction entirely (McKenzie et al., 2007; Viterbo et al., 1999). Dall'Amico et al.
(2011), Painter (2011) and Painter and Karra (2014) evaluated their respective Clapeyron equation
based cryosuction models in soil column freezing simulations and presented a good match between
simulations and laboratory experiments in total water content (liquid and ice). Recently, Stuurop
et al. (2021) applied an empirical expression, a physics-based expression, and no cryosuction in
simulating soil column freezing process. They compared the simulated results with observations
from laboratory experiments. This comparison demonstrated minor differences between empirical
and Clapeyron-based cryosuction expressions, but the simulation without cryosuction cannot
predict the distribution of total water content in a laboratory-scale soil column. To our knowledge,
there is still no literature showing the effect of cryosuction on plot-scale permafrost predictions.
Third, heat transport in process-rich models is described using an energy conservation equation,
mainly including heat conduction, latent heat exchange, and heat advection. From a continuum-
scale perspective, conductive heat transport is expressed in the form of a diffusive term base
on Fourier's law. Latent heat exchange accompanies phase change which alters the system enthalpy.
Advective heat transport describes the energy exchange caused by the flow of liquid water driven
by a hydraulic gradient (i.e., forced convection), which is expressed through an advective term in
energy balance equations. Additionally, other mechanisms that control heat transport, such as
water vapor movement, thermal dispersion, radiant energy, etc., are neglected by nearly all models
of permafrost and are not considered here. Among these heat transport mechanisms, it is
commonly recognized that heat conduction predominates heat transport in the subsurface (Nixon,
1975). However, there are also studies demonstrating the importance of advective heat transport
in permafrost hydrology through field observation analysis or modeling comparison. Such
situations where advective heat makes important contributions roughly fall into three categories.
The first centers on the development of taliks beneath lakes, ponds, snowmelt water and rainfall
induced runoff, or in the existence of supra-/sub-permafrost groundwater flow (e.g., Luethi et al.,
2017; McKenzie and Voss, 2013; Rowland et al., 2011). The second focuses on microtopographic
features that focus significant amount of water flux through small areas. This includes both low-
center ice wedge polygons associated with the formation of thermokarst ponds (e.g., Abolt et al.,
2020; Harp et al., 2021) and thermo-erosion gullies (e.g., Fortier et al., 2007; Godin et al., 2014).
In these cases, large, focused flows across small spatial scales allow advective heat transport to





dominate. The last category includes those studying the construction and maintenance of
infrastructure influenced by groundwater flow (e.g., Chen et al., 2020). Thus, these studies focus
on either location-specific or scale-limited problems. As McKenzie and Voss (2013) stated,
whether heat advection outweighs heat conduction depends on soil permeability, topography, and
groundwater availability. Relative to these special cases, we are more interested in to what extent
advective heat transport associated with liquid water flow contributes to permafrost hydrologic
change in a hillslope-scale or larger Arctic system.
To clarify the significant differences in model representations of permafrost, we investigate the
influence of including or not including these processes on permafrost change at plot-to-hillslope
scales. We take the advantage of the flexibility offered by ATS to express multiple options of
process representation to implement this study in numerical experiments. For ice density, we
compare simulations with and without differences in ice density relative to water density; for
cryosuction, we compare simulations using a Clapyron equation-based expression and excluding
the cryosuction effect; and for heat transport, we compare simulations including or neglecting
advective heat transport. All comparisons are carried out across a range of Arctic climate
conditions and soil properties from three different sites. Both 1D soil-column-scale and 2D
hillslope-scale models are considered, in which varying hillslope geometries (i.e.,
convergent/divergent hillslope) and aspects (i.e., north/south) are included. The aim of this study
is to provide permafrost hydrology modelers with crucial comparisons for better choosing a model
representation for a given study by formally considering the tradeoff between model complexity,
accuracy, and, at least for one code, performance.
**2 Theory**
The Advanced Terrestrial Simulator (ATS v1.2) (Coon et al., 2020) configured in permafrost mode
(Jan et al., 2018, 2020; Painter et al., 2016) was used to implement all numerical experiments in
this study. ATS is a process-rich code developed for simulating integrated surface and subsurface
hydrological processes, specifically capable of permafrost applications. It has been shown to
successfully compare to observations of seasonal soil freezing and thaw processes at different
scales. This includes 1D models of vertical energy transport typical of large-scale flatter regions
(Atchley et al., 2015), and 2D models admitting lateral flow and transport in Arctic fens (Sjöberg
et al., 2016),  and polygonal ground (Jan et al., 2020).





The permafrost configuration of ATS comprises coupled water flow and energy transfer within
variably saturated soils and at land surfaces, a surface energy balance model describing thermal
processes in snow, and a snow distribution module for surface microtopography (Painter et al.,
2016). The subsurface system solves a three-phase (liquid, ice, gas), two-component (water vapor,
air) Richards-type mass balance equation with Darcy's law and an advection-diffusion energy
balance equation. The surface system includes an overland flow model with diffusion wave
approximation, and an energy balance equation with an introduced temperature-dependent factor
describing the effect of surface water freezing. The subsurface system and surface system are
coupled through the continuity of pressure, temperature, and the corresponding fluxes by
incorporating the surface equations as boundary conditions of the subsurface equations (Coon et
al., 2020). The evolution of a snowpack and its effect on the surface energy balance is described
using an energy balance approach based on a subgrid model concept that includes all major heat
fluxes at the land surface. For a more detailed description of the permafrost configuration and
implementation in ATS, as well as key mathematical equations, the reader is referred to Painter et
al. (2016). Changes in this "most complex" model of permafrost hydrology are enabled by the
Arcos multiphysics library leveraged in ATS; this allows the precise model physics to be specified
and configured at runtime through the use of a dependency graph describing swappable
components in the model physics (Coon et al., 2016).
**2.1 Ice density**
The density of ice (kg/m$^3$) is represented as a Taylor series expansion in both temperature and
pressure:
$$\rho_i = [a + (b + c\Delta T) \times \Delta T] \times (1 + \alpha \Delta p) \tag{1}$$
and the density of liquid water (kg/m$^3$) is represented as:
$$\rho_l = [a + (b + (c + d\Delta T) \times \Delta T) \times \Delta T] \times (1 + \alpha \Delta p) \tag{2}$$
where $\Delta T = T - 273.15$, $\Delta p = p_l - 1e5$, $T$ and $p_l$ are temperature (K) and liquid pressure (>101325
Pa), respectively; and $a$, $b$, $c$, $d$, $\alpha$ are constant coefficients, listed in Table 1. Under conditions of
equal density, we assume $\rho_i = \rho_l$.

179                                       **Table 1 Coefficients in density of ice and liquid**

| | $a$ | $b$ | $c$ | $d$ | $\alpha$ |
|---|---|---|---|---|---|
| $\rho_i$ | 916.724 | -0.147143 | -2.38e-4 | – | 1.0e-10 |
| $\rho_l$ | 999.915 | 0.0416516 | -1.01e-2 | 2.06e-4 | 5.0e-10 |



## 2.2 Cryosuction

Painter and Karra (2014) proposed a constitutive relationship for phase partitioning of water in
frozen soils based on Clapeyron equation and Van Genuchten model (Van Genuchten, 1980):
$$s_l = \begin{cases} S_*(-\beta\rho_l L_f \vartheta), \vartheta < \vartheta_f \\ S_*(p_g - p_l), \vartheta \geq \vartheta_f \end{cases}, \quad \vartheta = \frac{T(K)-273.15}{273.15}, \quad \vartheta_f = -\frac{\psi_*(1-s_g)}{\beta L_f \rho_l}$$
$$s_i = 1 - s_l / S_*(p_g - p_l)$$
(3)

where $s_n$ is the saturation of $n$-phase and the subscripts $n = l$, i, g are liquid, ice, and gas phases,
respectively; $\beta$ is a coefficient; $L_f$ is the heat fusion of ice; $p_n$ ($n = l$, g) is the pressure of $n$-phase;
$S_*$ is the Van Genuchten model. This physically derived formulation can describe the change of
matric suction in the frozen zone due to the change of ice content, and thus has the capacity to
represent cryosuction.
Alternatively, to exclude the effect of cryosuction in this study, we used the Van Genuchten model
to determine the total water content, including liquid water and ice. The liquid water content is
achieved by an empirical relationship (soil-freezing characteristic curve) which describes that the
liquid water content only relates to temperature through an exponent function (McKenzie et al.,

193 2007).

$$s_l = s_r + (s_{sat} - s_r)\exp\left[-\left(\frac{T(K)-273.15}{\omega}\right)^2\right]$$
$$s_i = S_*(p_g - p_l) - s_l$$
(4)

where $s_r$, $s_{sat}$ are saturations of liquid water at residual and saturated conditions, respectively; $\omega$ is
a constant coefficient.

## 2.3 Advective heat transport

The energy conservation equation of the subsurface system is given by:
$$\frac{\partial}{\partial t}\left[\phi \sum_{n=l,i,g}(\rho_n s_n u_n) + (1-\phi)c_{v,soil}T\right] + \underbrace{\nabla \cdot (\rho_l h_l \mathbf{V}_l)}_{\text{advective heat}} \underbrace{-\nabla \cdot (\kappa_e \nabla T)}_{\text{conductive heat}} = Q_E$$
(5)

where $\phi$ is porosity; $u_n$ is the specific internal energy of phase ($n \in \{l, i, g\}$); $c_{v,soil}$ (J m$^{-3}$ K$^{-1}$) is
the volumetric heat capacity of the soil grains. The second and third terms represent the advective
and conductive heat transport in subsurface, in which $h_l$ (J/mol) is the specific enthalpy of liquid;
$\mathbf{V}_l$ (m/s) is the velocity vector of liquid water determined by Darcy's law; and $\kappa_e$ (W m$^{-1}$ K$^{-1}$) is
the effective thermal conductivity of the bulk material including soil, air, liquid water, and ice. $Q_E$
is the sum of all thermal energy sources (W/m$^3$).
Similarly, the energy balance equation of the surface system is:





$\frac{\partial}{\partial t}\{[\chi\rho_l u_l + (1-\chi)\rho_i u_i]\delta_w\} + \underbrace{\nabla \cdot (h_l\chi\rho_l\delta_w\mathbf{U}_w)}_{\text{advective heat}} \underbrace{-\nabla \cdot \{[\chi\kappa_l + (1-\chi)\kappa_i]\delta_w\nabla T\}}_{\text{conductive heat}} = Q_{\text{net}}$     (6)
in which $\chi$ is the unfrozen fraction determined by surface temperature; $\delta_w$ is ponded depth (m);
$\mathbf{U}_w$ (m/s) is the velocity vector of liquid water on the surface determined by the diffusion-wave
approximated St. Venant equations (Gottardi and Venutelli, 1993) and Manning equation
(Wasantha Lal, 1998); $\kappa_n$ (W m$^{-1}$ K$^{-1}$) is the thermal conductivity of $n$-phase ($n = l, i$); $Q_{\text{net}}$ (W/m$^3$)
is the net thermal energy into and out of surface, including that from solar radiation, rain and snow
melt, water loss by evaporation and to the subsurface, and conductive and advected heat transport
to/from the subsurface. The second and third terms represent the (lateral) advective and conductive
heat transport that occur across the land surface.
**3 Methods**
To evaluate the impact of representation of ice density, cryosuction, and advective heat transport
in permafrost modeling under different climate conditions and soil properties, we selected three
sites for their variance in climactic condition: Utqiagvik (Barrow Environmental Observatory,
71.3225º N, 156.6231º W), the headwaters of the Sagavanirktok (Sag) River (68.251º N, 149.092º
W), and the Teller Road Mile Marker 27 site on the Seward Peninsula (64.73º N, 165.95º W) in
Alaska. The simulated hydrological outputs for each site are compared in both column and
hillslope scenarios. Column scenarios represent expansive flat regions typical of the Arctic coastal
plains dominated by vertical infiltration and heat transport, and hillslope scenarios are
representative of the headwater, hilly terrain typical of the more inland permafrost.
In hillslope scenarios, hillslopes with northern and southern aspects are considered to investigate
physics representation comparisons under the same climate and soil condition (i.e., at a given site)
but different solar radiation incidence. Furthermore, hillslopes with both convergent and divergent
geometries are included to compare the sensitivity of simulated discharge on process
representation. These scenarios can incorporate many types of Arctic systems at the described plot-
to-regional scales, but explicitly ignore the effects of microtopography or other local-scale
focusing mechanisms such as water tracts or thermo-erosion gullies. The objective is to reach a
conclusion on the influence of the three physics representation that can be widely applicable in
many Arctic systems.
**3.1 Field data description**



For each site, data used in each simulation comprises meteorological forcing datasets for the period
2011-2020, averaged wind speed, and soil properties.
Meterological forcing datasets are taken from the Daymet version 4 dataset (Thornton et al., 2020),
which provides observation-based, daily averaged weather variables through statistical modeling
techniques at 1 km spatial resolution (Thornton et al., 2021). Variables that are used in simulations
include daily average air temperature (calculated as the mean of Daymet's daily minimum and
maximum values), relative humidity (calculated from air temperature and Daymet's vapor
pressure), incoming shortwave radiation (W/m$^2$) (calculated as a product of Daymet's daylit
incoming radiation and daylength), and total precipitation (m/s), which is split into snow and rain
based upon the air temperature. Figure 1 illustrates the precipitation of rain, snow, and air
temperature in the three sites from 2011 to 2020, where the points represent the corresponding
averaged values per year. In terms of the forcing conditions, the annual rainfall of the Sag and
Teller sites range between 20 and 40 mm/d over the ten years, more than twice the rainfall typical
of the Barrow site. In addition, Sag has a significantly larger amount of snow every year that is
over double of that at the Teller site and almost five times larger compared to the Barrow site. For
the air temperature, Sag and Barrow are similar and colder than Teller by 7-8 degrees. In general,
the Barrow site is dry and cold, the Sag site is wet and cold, and the Teller site is wet and warm.

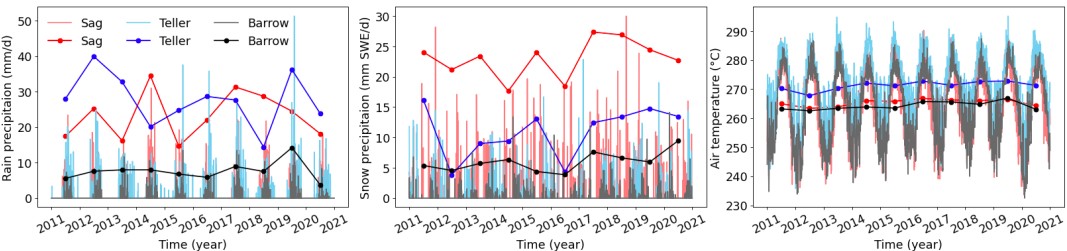


**Figure 1 Precipitation and air temperature of site Barrow, Sag, and Teller from year 2011 to 2020**
In addition to the time series of forcing data from Daymet, we used an average wind speed for
each site. For Barrow and Teller, the average wind speed was estimated from the measurement
taken by the Next-Generation Ecosystem Experiments (NGEE) Arctic project. At Barrow, the
measurement was taken at area A (71.2815º N, 156.6108 º W) at the height of 1.3 meters above
surface (Hinzman et al., 2014). At Teller, the measurement at 3.8 m above the surface of a lower
level of the watershed (Busey et al., 2017) was used. For Sag, the average wind speed was
estimated based on the measurement at the Toolik Lake field site (near to Sag River) at the height



of 3.1 m above surface, which is accessible through the National Ecological Observatory Network
(NEON, 2021).
The soil properties of Barrow, Sag, and Teller, including porosity, permeability, Van Genuchten
parameters, and thermal conductivity parameters, were chosen from previous modeling studies at
these sites (Atchley et al., 2015; Jafarov et al., 2018; O'Connor et al., 2020), see (Table 2). Roughly,
the soil profile of each site is composed of two materials: the top organic-rich layer comprising
mosses, peats, and other organic rich soils measuring approximately 10-30 cm thick, and the
principal mineral soil. There is minor difference in thermal conductivity parameters among the
three sites, and soil permeability is also at the same order of magnitude. The soil-water
characteristic curve (SWCC) of the principal mineral soil of Barrow, Sag, and Teller, shown in
Figure 2, indicates that the soil property between Barrow and Teller is relatively similar, while Sag
differs from the other two with a relatively flat SWCC.

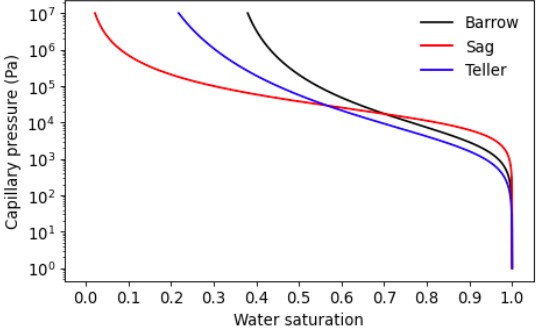


**Figure 2 Soil-water characteristic curve (SWCC) of soil in Barrow, Sag, and Teller**
Usually, at the hillslope scale, the thickness of organic layers of a watershed varies from the toe-
slope, through a steeper mid-hill, up to the flat top. Typically, thicker organic layers may exist at
the top and bottom compared to the mid-hillslope. The low thermal conductivity of organic layers
can impede the heat transport between the air and the underlying mineral soil, resulting in varying
thaw depth (or permafrost table depth) along a hillslope, which has been observed at the site Teller
(Jafarov et al., 2018). In this paper, hillslope meshes were constructed following this observation
so that the organic layers are thicker at the top and bottom of a hillslope, as described in the next
section.
**3.2 Mesh design and material properties**
The comparison of different physics representations was conducted in both column and hillslope





scenarios.
The column model was designed as a one-dimensional, 50 m deep domain. The column domain
was discretized into 78 cells with gradually increasing cell thickness, starting from 2 cm at the soil
surface to 2 m at the bottom of the domain. We assigned different material properties to the cells
to represent different soil layers. A column domain is divided into three layers, and the thickness
of each layer was designed differently among the three sites according to geological observations
(Jan et al., 2020; O'Connor et al., 2020; NGEE-Arctic). Specifically, from top to bottom, the three
layers of the Barrow soil column are 2 cm-thick moss, 8 cm-thick peat, and mineral; for Teller, the
soil column consists of a 4 cm moss layer, a 22 cm peat layer, and mineral; and the three layers of
the Sag soil column are acrotelm, catotelm, with thickness of 10 cm and 14 cm, respectively, and
the remainder mineral. The soil properties of each layer at three sites are listed in Table 2.
**Table 2 Soil properties of three soil layers of all sites used in this paper**

| Site | Barrow | | | Sag | | | Teller | | |
|---|---|---|---|---|---|---|---|---|---|
| Layers | moss | peat | mineral | acrotelm | catotelm | mineral | moss | peat | mineral |
| Porosity | 0.9 | 0.876 | 0.596 | 0.878 | 0.796 | 0.457 | 0.9 | 0.55 | 0.45 |
| Permeability (m$^2$) | 1.7e-11 | 9.38e-12 | 6e-13 | 2.64e-10 | 9.63e-12 | 3.98e-13 | 5e-11 | 5e-12 | 2e-13 |
| VG $\alpha$ (Pa$^{-1}$) | 2.3e-3 | 9.5e-4 | 3.3e-4 | 7.93e-4 | 1.75e-4 | 8.06e-5 | 2.35e-3 | 2.93e-4 | 5.45e-4 |
| VG n | 1.38 | 1.44 | 1.33 | 1.405 | 1.566 | 1.571 | 1.38 | 1.269 | 1.236 |
| Residual saturation | 0.056 | 0.388 | 0.334 | 0.0073 | 0.0662 | 0. | 0.1 | 0. | 0.1 |
| Thermal conductivity, unfrozen (Wm$^{-1}$K$^{-1}$) | 0.446 | 0.427 | 0.788 | 0.519 | 0.630 | 1.309 | 0.57 | 0.67 | 1 |
| Thermal conductivity, dry (Wm$^{-1}$K$^{-1}$) | 0.024 | 0.025 | 0.104 | 0.066 | 0.086 | 0.265 | 0.07 | 0.07 | 0.29 |


In the hillslope scenario, we designed the mesh based on observations at Teller to represent a
generalized, varying-thickness low Arctic hillslope. A hillslope mesh was created first by
generating a pseudo-2D surface mesh with 50 cells and then extruding the 2D mesh downward by
50 m. The pseudo-2D surface was designed in a trapezoidal shape with a single, variable-width
cell in the cross-slope direction to represent convergent/divergent hillslopes, the short and long
sides of which are 200 m and 800 m, respectively (see Figure 3). Vertically, from surface
downward, the grid size distribution was the same as the column mesh for each site. The domain
is also composed of three layers, same as the column, while the numbers of cells representing each
soil layer (i.e., soil layer thickness) are different along the hillslope. The thickness distribution of
the first two layers of each site is shown in Table 3. The third layer of a hillslope for all sites is the
principal mineral soil. Additionally, hillslope meshes with different aspects (i.e., north, south) were





also created.

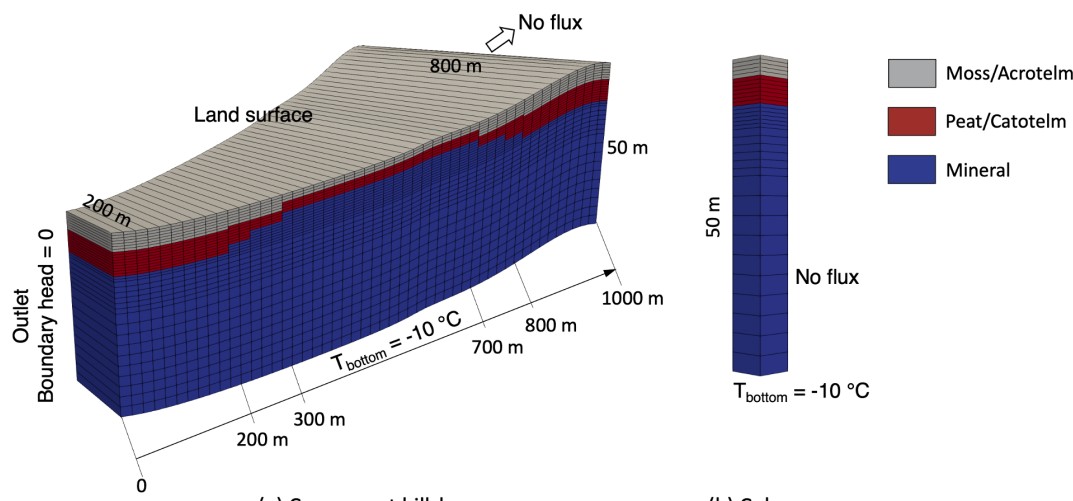

**Figure 3 Schematic domain mesh and soil layer partition: (a) example of a convergent hillslope domain, (b)**
**column domain.**

**Table 3 Thickness distribution of the organic layers along hillslope for each site**

| Site | Horizontal $x$ range (m) | Barrow layer thickness (cm) | Sag layer thickness (cm) | Teller layer thickness (cm) |
|---|---|---|---|---|
| Layer 1 Moss/Acrotelm | 0 ~ 200 | 2 | 14 | 8 |
| | 300 ~ 700 | 2 | 6 | 4 |
| | 800 ~ 1000 | 2 | 14 | 8 |
| Layer 2 Peat/Catotelm | 0 ~ 200 | 12 | 18 | 22 |
| | 300 ~ 700 | 6 | 8 | 22 |
| | 800 ~ 1000 | 12 | 18 | 22 |

**3.3 Model setup**
To study how the representations of the three physical processes (i.e., ice density, cryosuction, and
advective heat transport) affect simulated hydrological outputs at different scales and hillslope
topography features, and under various forcing and soil conditions, 62 model simulations were
conducted, summarized in Table 4. To examine the validity of the assumption of equal density
between ice and liquid, we included cryosuction and advective heat transport in models. To
investigate the role of cryosuction in permafrost modeling, we used different density, while
neglecting advective heat transport to decrease the computation cost. Note that neglecting
advective heat transport in these runs can reduce the effect of cryosuction on simulation predictions,
as cryosuction moves water which would itself advect energy. To compare the difference between
neglecting and including heat advection, we used different density expressions for ice and liquid,





and included cryosuction. Particularly, in order to understand the impact of advective heat
transport on permafrost process when soil is at its wettest, we designed two extreme cases under
the warm, wet conditions of the Teller site in which soil evaporation was artificially reduced.
These runs were designed to maximize water flux and therefore maximize the potential for
advective heat transport to affect predictions.
**Table 4 Ensemble of models designed in this study**

| To compare | Site | Scale | Geometry | Aspect | Remark |
|---|---|---|---|---|---|
| • $\rho_i \neq \rho_l$, Eq. (1)<br><br>• $\rho_i = \rho_l$, Eq. (2) | Barrow<br>Sag<br>Teller | column | – | – | • heat advection<br><br>• cryosuction |
|  |  | hillslope | convergent | north |  |
|  |  |  |  | south |  |
|  |  |  | divergent | north |  |
|  |  |  |  | south |  |
| • Include heat transport<br><br>• Neglect heat transport | Barrow<br>Sag<br>Teller | column | – | – | • $\rho_i \neq \rho_l$<br><br>• cryosuction |
|  |  | hillslope | convergent | north |  |
|  |  |  |  | south |  |
|  |  |  | divergent | north |  |
|  |  |  |  | south |  |
|  | Extreme case, Teller | hillslope | convergent | north | • reduced evaporation |
| • Include cryosuction<br><br>• Neglect cryosuction | Barrow<br>Sag<br>Teller | column | – | – | • $\rho_i \neq \rho_l$<br><br>• no heat advection |
|  |  | hillslope | convergent | north |  |
|  |  |  |  | south |  |
|  |  |  | divergent | north |  |
|  |  |  |  | south |  |


Prior to simulating all cases, two steps of initialization are carried out for each site. First, a column
model with a given initial water table depth and above-0 ℃ temperature was frozen by setting the
bottom temperature at a constant value of -10 ℃ until a steady-state frozen soil column is formed.
The initial water table depth is chosen to ensure that the frozen column's water table, after
accounting for expansion of ice, is just below the soil surface. The pressure and temperature
profiles of the frozen column were used as the initial conditions of the second step initialization.
Before proceeding, the observed forcing data (period of 2011-2020) was averaged across the years
to form a one-year, "typical" forcing year, which was then repeated 10 times. Using this typical
forcing data and the solutions of the first step, we solved the column model in a transient solution,
calculating an annual cyclic steady-state and obtaining the pressure and temperature fields at the
end of the 10th year. The final state was then used as initial condition in formal simulations listed
in Table 4. The temperature at bottom was constant at -10 ℃.
**3.4 Evaluation metrics**



To fully assess the effect of representation of ice density, advective heat transport, and cryosuction
in permafrost hydrology modeling, we used root mean squared error (RMSE) and normalized
Nash–Sutcliffe efficiency (NNSE) as performance metrics. RMSE has the same dimension with
the corresponding variables, which can be used to evaluate the average absolute deviation from a
benchmark, defined by:
$\text{RMSE} = \sqrt{\frac{\sum_{t=1}^{N}(x_t - y_t)^2}{N}}$         (7)
where $x_t$ and $y_t$ are the two modeled datasets to compare from the initial time point ($t = 1$) to the
end ($t = N$).
NNSE is a normalized dimensionless metric describing the relative relationship between an
estimation and a reference, which is oftentimes used for evaluating hydrological models.
$\text{NNSE} = 1 / \left(1 + \frac{\sum_{t=1}^{N}(x_t - y_t)^2}{\sum_{t=1}^{N}(x_t - \bar{x})^2}\right)$         (8)
where the modeled results $x_t$ (obtained without physics simplification) is considered as the
benchmark, and $\bar{x}$ is the mean value of the benchmark. NNSE approaching to 1 indicates perfect
correspondence between two observations.
In addition, we also used normalized mean absolute error (MAE) to quantify the percentage change
of results obtained with simplified physics relative to full physical representations (see Section

4.4).

$\text{Normalized MAE} = \frac{\sqrt{\sum_{t=1}^{N}|x_t - y_t|/N}}{\text{normalizing reference}} \times 100\%$         (9)
Two normalizing references were selected considering different model output variables. For
instance, in terms of temperature and saturation which fluctuate between two non-zero values, the
annually averaged variation range was chosen as the reference.
$\text{Normalizing reference} = \frac{\sum_{\text{year}=1}^{\text{num of years}}(\text{maximum} - \text{minimum})}{\text{number of years}}$
For variables with zero as the smallest value, such as evaporation, discharge, and thaw depth, the
corresponding average value was selected as the reference.
**4 Results**
This section compares simulated outputs over the period of 2011-2020 for the three physics under
different simulating conditions. We focus on the impact on integrated variables, such as
evaporation, discharge, averaged thaw depth, and depth-dependent variables, such as temperature,




and total water saturation (ice and liquid). For hillslope models, we chose five surface locations
according to the slope geometry to collect simulated data, which were then averaged to obtain a
single outcome for each variable of interest.
**4.1 Ice density**
To evaluate the representation of ice density on permafrost process simulation, we compared
evaporation, discharge, thaw depth, and total water saturation between simulations using equal and
different ice density expressions. Figure 4 and Figure 5 show an example of the comparison under
conditions of Sag at column and hillslope scale, respectively. Results are compared in both time
series and correlation.
Generally, at both column and hillslope scale, assuming equal density between ice and liquid has
minor impacts on evaporation, discharge, and thaw depth over the 10-year simulation, except at a
few deviated points as shown in the correlation figures. According to column-based models, the
RMSEs of evaporation, discharge, and thaw depth are 0.101 mm/d, 0.001 $m^3$/d, and 1.648 cm,
respectively, one order of magnitude smaller than the corresponding variable values. At hillslope
scale, see Figure 5, the south-facing divergent hillslope is selected to show modeling comparison
on evaporation and thaw depth, in that they are potentially mostly affected when a hillslope has a
south orientation and divergent geometry. Likewise, the north-facing convergent hillslope is
chosen to compare discharge and water saturation from simulations with different density
expression. Even then, RMSEs of the three variables are 0.064 mm/d, 111.073 $m^3$/d, and 0.825 cm,
respectively, two orders of magnitude smaller than the corresponding variable values at hillslope
scale. Besides, NNSEs of the three variables output from both column and hillslope simulation are
over 0.9, approaching 1 especially at the hillslope scale. Therefore, all indicate good performance
of equal ice-liquid density assumption in predicting integrated variables and thaw depth. By
comparison, the estimation of water saturation is relatively affected by the density assumption
during cold seasons within a year, as shown by Figure 4 (d) and Figure 5 (d). This is reasonable in
that when water mainly exists in the form of ice, equal ice-liquid density assumption will
overestimate the water content.





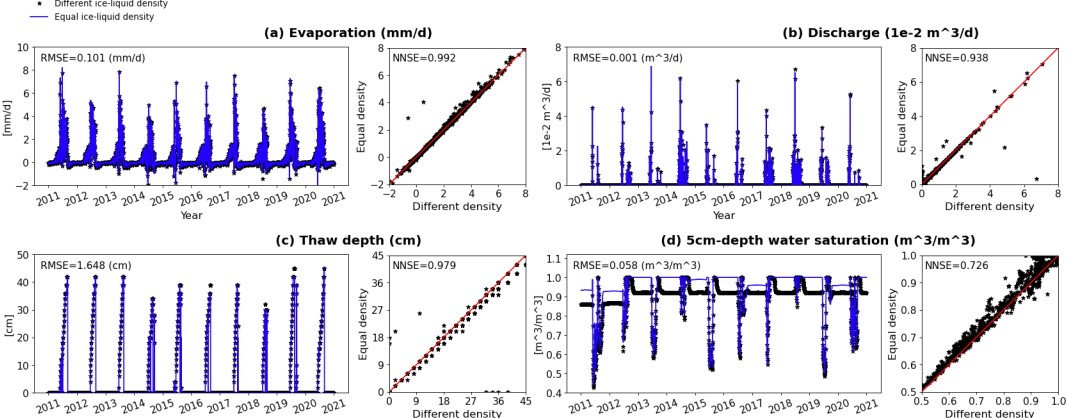

**Figure 4 Comparison of column simulations between different and equal ice-liquid density under conditions of Sag, in (a) evaporation, (b) discharge, (c) thaw depth, and (d) water saturation at 5 cm beneath surface from year 2011 to 2020.**

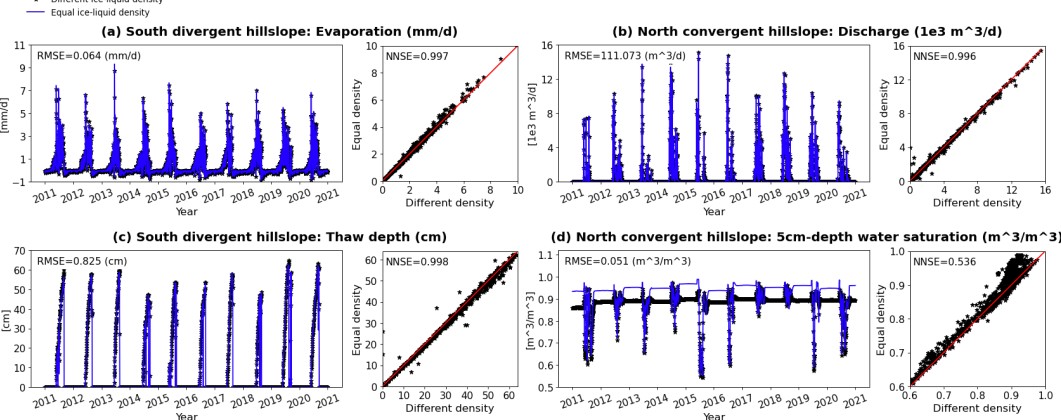

**Figure 5 Comparison of hillslope simulations between using different and equal ice-liquid density under conditions of Sag, in (a) evaporation, (b) discharge, (c) thaw depth, and (d) water saturation at 5 cm beneath surface from year 2011 to 2020.**

In addition, we investigated how much the assumption of equal ice-liquid density can affect simulation time at hillslope scale. Using 10-year simulations with real ice density as references, the percentage change of time consumed after applying equal ice-liquid density was calculated and displayed in Figure 6. Overall, under the density assumption, it may take less time (positive percentage), but no more than 25% and on average lower than 10%. However, it may also increase computational time (negative percentage) mainly under wet conditions, such as at Sag and Teller. Thus, given a long-period large-scale modeling of permafrost freezing and thaw process, there is no consistent conclusion on whether equal ice-liquid density can ease computational cost. It




depends on both the weather conditions and soil properties.

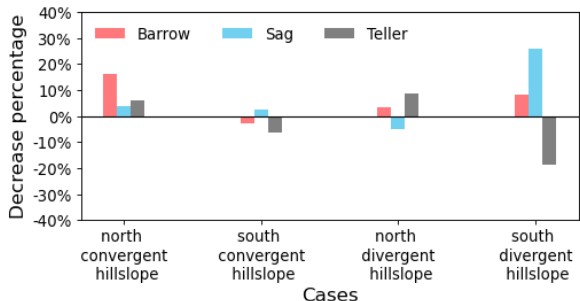

**Figure 6 Decreased percentage of simulation time under the assumption of equal ice-liquid density compared**
**to the real ice density representation for all hillslope scale simulations.**
**4.2 Cryosuction**
To evaluate the effect of cryosuction on permafrost process predictions, we compared evaporation,
discharge, thaw depth, total water saturation, and temperature obtained through simulations
including and neglecting cryosuction. Figure 7 through Figure 9 illustrate column-scale
comparisons of these variables under conditions at three sites (Barrow, Sag, and Teller). Figure 7
presents the effect of excluding cryosuction on evaporation and discharge. RMSE of evaporation
from the three sites ranges between 0.25 mm/d and 0.35 mm/d, still one order of magnitude smaller
than the common evaporation rate. Evaporation NNSEs of the three sites are around 0.9. For
discharge, RMSEs are also one order of magnitude smaller than the average, whereas NNSEs fall
between 0.6 and 0.9. Generally, cryosuction plays a more important role in predicting discharge
compared to evaporation, especially under warm and wet climate conditions, such as Teller.



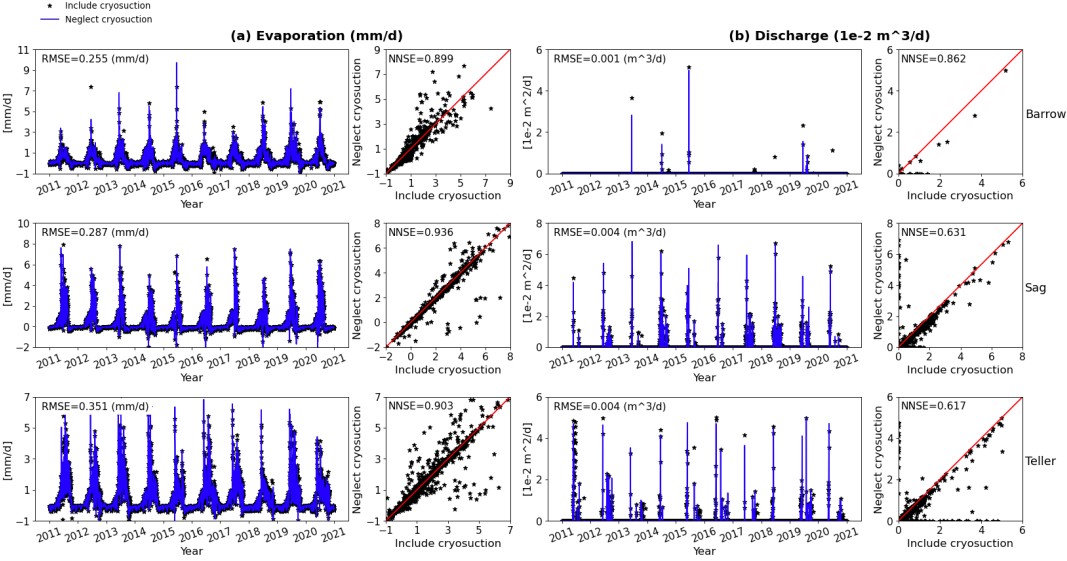

**Figure 7 Comparison of column simulations between including and neglecting cryosuction under conditions of Barrow, Sag, and Teller, in (a) evaporation, (b) discharge.**

Figure 8 shows the effect of cryosuction on column-scale simulated thaw depth and total water saturation at 5 cm beneath surface. RMSEs of thaw depth range from 3 cm to 8 cm. Though still one order of magnitude smaller than the average annual thaw depth, the estimation error due to neglecting cryosuction is obvious in summer, especially at areas with cold temperature and low rainfall like Barrow. By comparison, at Teller, where the largest thaw depth is over double of Barrow and Sag due to its higher temperature, soil cryosuction does not essentially affect thaw depth compared to the other two sites. Similarly, for the total water saturation, at Barrow, the effect of cryosuction is more clearly observed, not only during cold seasons as observed for density representation (section 4.1), but also in summers.





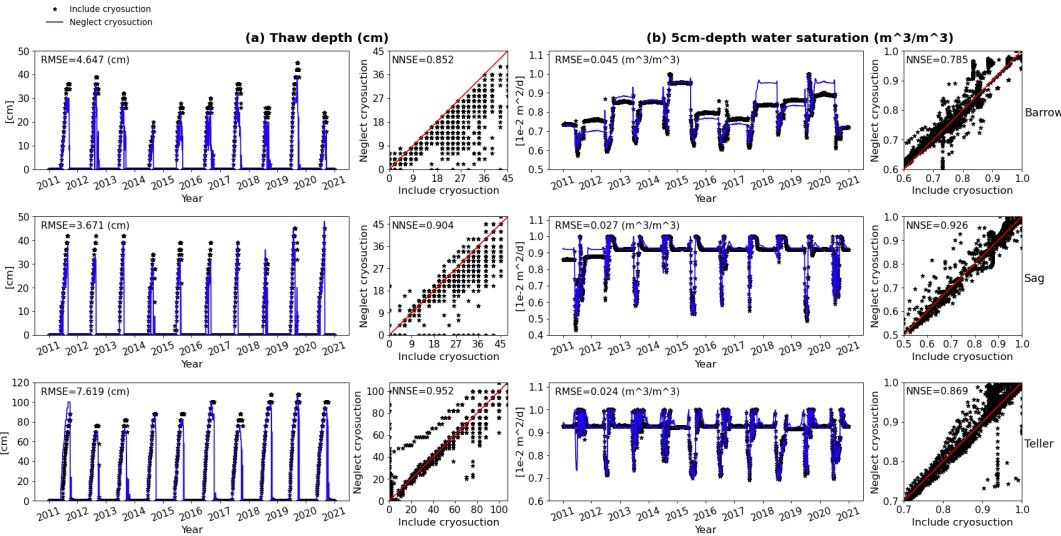

**Figure 8 Comparison of column simulations between including and neglecting cryosuction under conditions of Barrow, Sag, and Teller, in (a) thaw depth, (b) water saturation at 5 cm beneath surface.**

Finally, we also compared soil temperature obtained from models with or without cryosuction included; see Figure 9. Surface temperature is little affected by cryosuction, except at the Sag site, where the surface temperature is overestimated during winter. At 1 m depth, soil temperature of Barrow is slightly changed in summer due to neglecting cryosuction. At both Sag and Teller, the fluctuation range of temperature at 1 m beneath land surface is underestimated if cryosuction effect is not considered, especially at Sag, NNSE decreases to 0.6 approximately.

Therefore, from Figure 7 to Figure 9, neglecting cryosuction effect at column scale simulation has less impact on integrated hydrological variables, but will cause significant difference when estimating thaw depth and location-based variables. The difference among variables varies under different climate conditions. Influence on integrated variables, such as evaporation and discharge, are more obviously under warm and wet conditions (Teller); thaw depth and water saturation are affected more under cold and low-rainfall conditions (Barrow); and soil temperature tends to be influenced greater under cold and high precipitation (rain and snow) conditions (Sag).



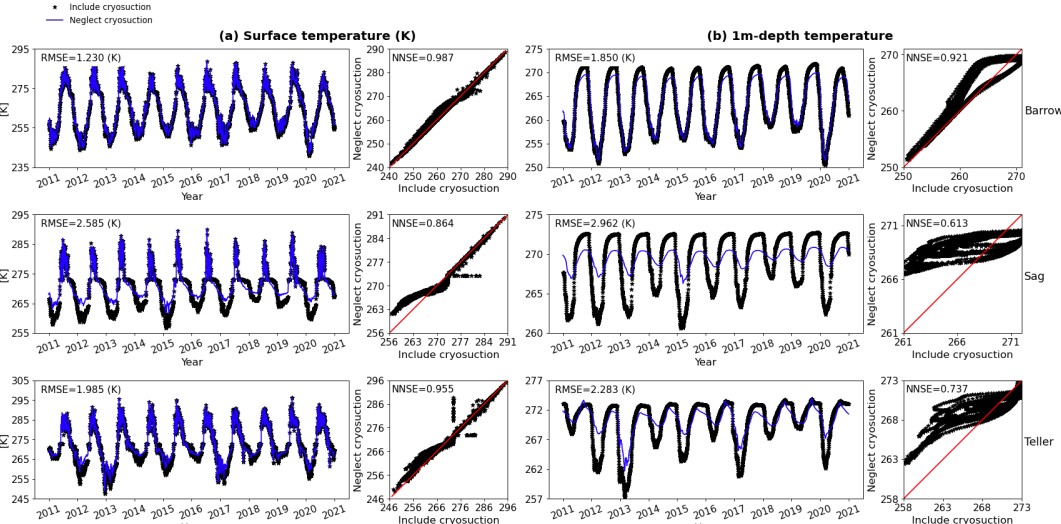

**Figure 9 Comparison of column simulations between including and neglecting cryosuction under conditions of Barrow, Sag, and Teller, in (a) surface temperature, (b) temperature at 1 m beneath surface.**

Neglecting soil cryosuction has a similar impact on hydrological outputs in hillslope scale models. Figure 10 shows the comparison of the variables discussed above under the Sag climate. Evaporation, thaw depth, and temperature are presented based on south-facing divergent hillslope models, while discharge and water saturation are from hillslope models with north-facing convergent geometry. In general, neglecting soil cryosuction has a smaller effect on integrated variables (evaporation and discharge) compared with other pointwise variables. Though thaw depth presents a high NNSE, approximately 0.94, and low RMSE, about 4.5 cm compared to the average, indicating a good match between models considered and excluded cryosuction, the estimation error during summer may reach as high as 10 cm, particularly from 2011 to 2017, as shown in Figure 10 (c). Obvious errors in water saturation and temperature, similar with column-scale models, occur almost annually with respect to extrema during winter and summer. Overall, compared to column-scale models, differences in evaporation, discharge, thaw depth, and surface temperature due to neglecting cryosuction effect are relatively reduced at hillslope scale if comparing NNSEs (Table 5). Localized subsurface variables, such as water saturation and 1m-depth soil temperature, show increased errors from column to hillslope scale models, which is primarily caused by lateral flux exchange captured by hillslope modeling.



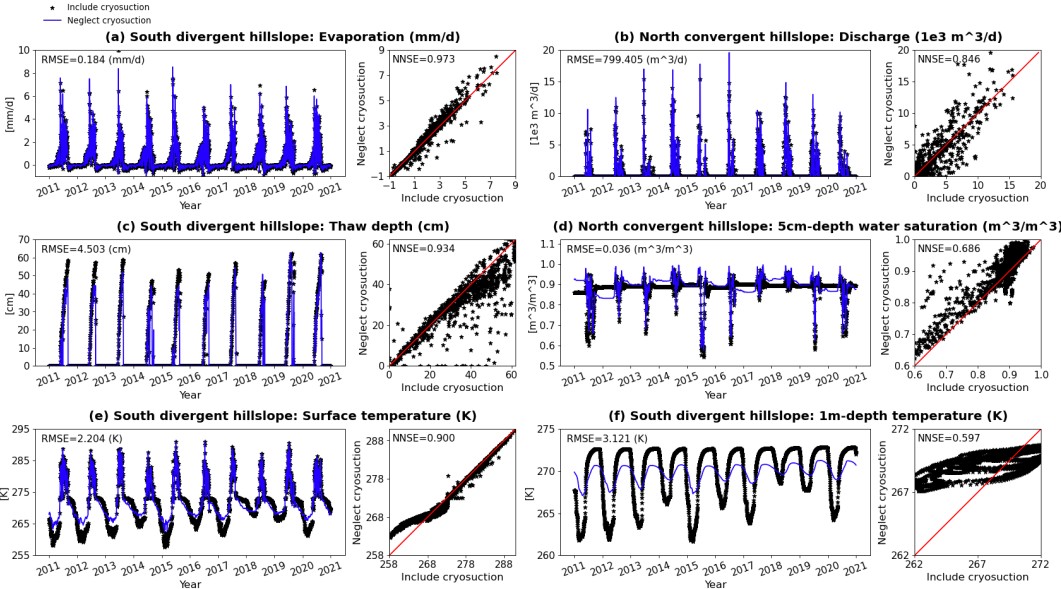

**Figure 10 Comparison of hillslope simulations between including and neglecting cryosuction under conditions of Sag, in (a) evaporation, (b) discharge, (c) thaw depth, (d) water saturation at 5 cm beneath surface, (e) surface temperature, (f) temperature at 1 m beneath surface.**

**Table 5 NNSE of outputs from column and hillslope models under conditions of Sag shown in Figure 7 through Figure 10**

| Scale | Evaporation (mm/d) | Discharge (m³/d) | Thaw depth (cm) | 5cm-depth water saturation (-) | Surface temperature (K) | 1m-depth temperature (K) |
|---|---|---|---|---|---|---|
| Column | 0.936 | 0.631 | 0.904 | 0.926 | 0.864 | 0.613 |
| Hillslope | 0.973 | 0.846 | 0.934 | 0.686 | 0.900 | 0.597 |

## 4.3 Advective heat transport

This section evaluates the performance of advective heat transport in modeling permafrost process. As above, we investigated the influence of neglecting heat advection on evaporation, discharge, thaw depth, total water saturation, and temperature. Overall, heat advection does not play a vital role in a normal Arctic system after comparing all hydrological outputs from models with different heat transport representations. Comparisons based on column-scale and hillslope-scale models are not shown here (see Supplement); instead, the extreme case under conditions of Teller is presented (Figure 11). Teller is abundant in rainfall over the period of 2011-2020 (Figure 1). In the extreme case, evaporation was reduced factitiously to almost a quarter of the original value (see Figure 7 (a) at Teller and Figure 11 (a)) for the purpose of increasing water flow rates. For instance, discharge has quadrupled after adjusting evaporation by comparing Figure 11 (b) and Figure 7 (b)



at Teller. This specific scenario is chosen to maximize the potential effect of advective heat
transport in a hillslope-scale Arctic system. Figure 11 illustrates comparisons on all outputs
mentioned above from hillslope models without heat advection and with full thermal
representation. Apparently, all RMSEs are extremely small, at least two orders of magnitude lower
than the corresponding variable average. Almost all NNSEs are approximately one, even for thaw
depth, localized water saturation, and temperature. Therefore, for most Arctic systems at this scale,
it is reasonable to neglect advective heat transport.

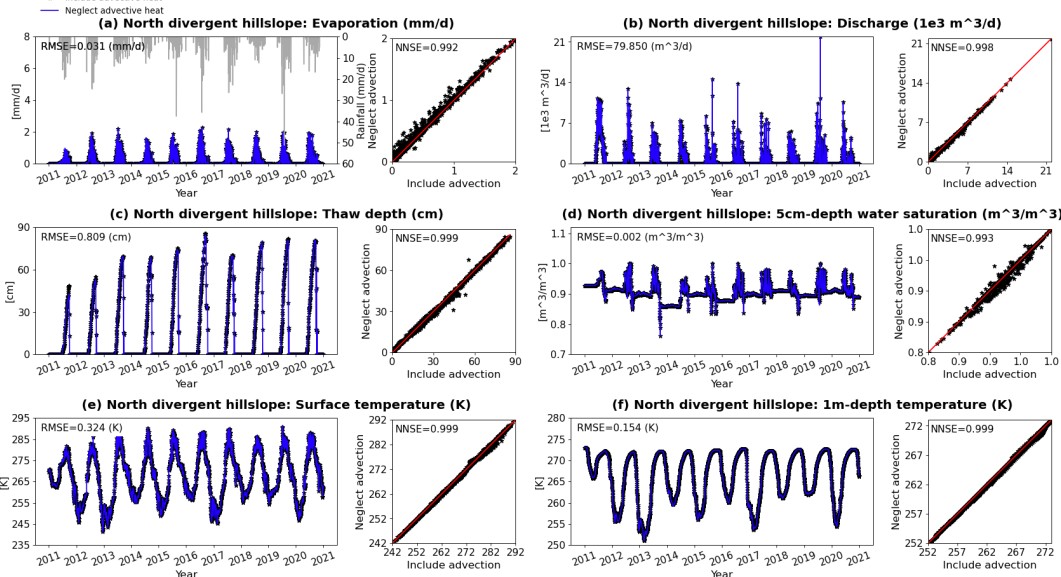

**Figure 11 Comparison of hillslope simulations between including and neglecting advective heat transport**
**under extreme conditions of Teller, in (a) evaporation, (b) discharge, (c) thaw depth, (d) water saturation at 5**
**cm beneath surface, (e) surface temperature, (f) temperature at 1 m beneath surface.**
In addition to simulated results, we also compared simulation times in percentage change between
hillslope models neglecting and including heat advection. ATS uses Algebraic Multigrid method
as preconditioner for solving, which has a relatively deficient performance in dealing with
hyperbolic equations. Thus, incorporating advective heat transport will aggravate computational
cost, particularly in case of both large spatial and temporal scale. Figure 12 shows the relative
percentage reduction in computational time for 10-year simulations after excluding heat advection
in both surface and subsurface thermal flux. It drops by 70% ~ 80% under wet conditions (e.g.,
Sag and Teller) and 40% ~ 60% under dry conditions (e.g., Barrow). Hence, neglecting advective
heat transport considerably improves the performance of large spatial-temporal permafrost





hydrology simulations.

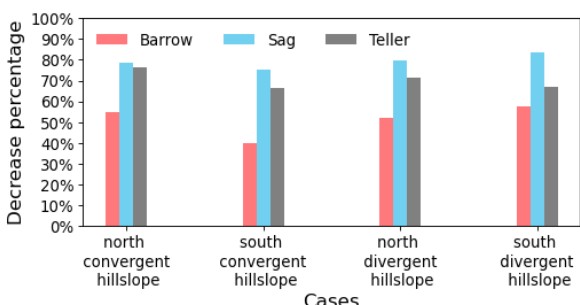

**Figure 12 Decreased percentage of simulation time after neglecting heat advection compared to full thermal**
**representation for all hillslope scale simulations.**
**4.4 Comprehensive comparison**
In the above three sections, we discussed time-series simulation comparisons. This section will
analyze the effect of equal ice-liquid density, neglecting cryosuction, and neglecting heat
advection on permafrost modeling outputs from holistic, average perspectives.
First, we extracted NNSEs of all variables obtained from all comparing models for qualitative
analysis. Table 6 shows an example based on column-scale models under conditions of three
different sites. Red numbers highlight the obviously reduced NNSEs of one or two processes
among the three for each variable. Overall, neglecting advective heat transport has the least
influence on model outputs. Equal ice-liquid density primarily affects saturation and has less effect
on other variables. Excluding soil cryosuction makes the greatest impact on almost all variables,
especially in a relatively wet environment. Among the variables, evaporation and surface
temperature are less affected by the three physical process representations, while location-based
water saturation is most affected.
**Table 6 A summary of NNSEs of variables obtained through column model comparison**

| Variables | Barrow | | | Sag | | | Teller | | |
|---|---|---|---|---|---|---|---|---|---|
| | *Heat advection* | *Ice density* | *Cryosuction* | *Heat advection* | *Ice density* | *Cryosuction* | *Heat advection* | *Ice density* | *Cryosuction* |
| Evaporation | 0.9971 | 0.9942 | 0.8991 | 0.9926 | 0.9917 | 0.9365 | 0.9989 | 0.9958 | 0.9033 |
| Discharge | 0.9235 | 0.6282 | 0.8615 | 0.9962 | 0.9377 | 0.6305 | 0.9854 | 0.9874 | 0.6175 |
| Thaw depth | 0.9970 | 0.9961 | 0.8517 | 0.9910 | 0.9791 | 0.9036 | 0.9969 | 0.9887 | 0.9524 |
| 5cm-depth $s_w$ | 0.9959 | 0.9335 | 0.7851 | 0.9916 | 0.7260 | 0.9260 | 0.9979 | 0.5618 | 0.8690 |
| 40cm-depth $s_w$ | 0.9932 | 0.0221 | 0.2130 | 0.9951 | 0.0622 | 0.3111 | 0.9990 | 0.2807 | 0.8498 |
| Surface $T$ | 0.9999 | 0.9999 | 0.9871 | 0.9993 | 0.9990 | 0.8642 | 0.9999 | 0.9996 | 0.9554 |
| 1m-depth $T$ | 0.9999 | 0.9999 | 0.9207 | 0.9997 | 0.9996 | 0.6127 | 0.9997 | 0.9991 | 0.7366 |



* $s_w$ and $T$ in Table 6 are water saturation and temperature, respectively.
Furthermore, to compare across the physics quantitively, we calculated the mean absolute error
(MAE) for each variable of interest over the simulation period of 2011-2020. For evaporation,
discharge, and thaw depth, the MAEs are normalized by the corresponding variable average
(numbers in Figure 13 (a)); for water saturation and temperature, the MAEs are normalized by
their average annual fluctuation range (numbers in Figure 13 (b)). All normalized MAEs are
presented in percentage, displayed in Figure 13 according to column- and hillslope-scale (e.g.,
south-facing convergent hillslope) models under three different climate conditions. Hillslope-scale
model output under conditions of Barrow is not shown in that flat land occupies a majority of the
area. A larger normalized MAE percentage indicates greater impact on the variable resulted from
a physical process.
From the perspective of 10-year average, in general, each physical process of Arctic system
discussed in this paper presents a similar impact on variables between column and hillslope scales.
Under climate and soil conditions of three different sites, neglecting cryosuction in permafrost
modeling leads to the greatest influence on hydrological prediction amongst the three physical
assumptions. As seen in Figure 13 (a), it will result in 10% ~ 20% deviation in evaporation, 50%
~ 60% in discharge, and 10% ~ 30% in thaw depth. Evaporation is the least affected among the
three variables. Discharge is more affected in regions with abundant rainfall (Teller), while in
regions with less precipitation, evaporation and thaw depth are relatively affected (Barrow). By
comparison, assuming equal ice-liquid density and neglecting advective heat transport may only
cause 10% and 5% or even much lower error, respectively, in reference to the annual average of a
variable. Specially in Barrow, models utilizing the same ice and liquid densities and ignoring
advective heat seem to make an obvious impact on discharge, whereas this also results from its
extremely low discharge (Figure 7 (b)).
Figure 13 (b) illustrates the normalized MAEs of water saturation at 5 cm beneath surface, as well
as temperature at surface and 1 m depth. The assumption of equal ice-liquid density primarily
affects the estimation of water saturation profile. It can lead to about 5% ~ 15% error relative to
the annual change range, and the error percentage tends to slightly decrease when applying
hillslope-scale models due to the inclusion of lateral flow. Apart from this, neglecting soil
cryosuction still makes the largest impact. Surface temperature is the least affected variable among
all these model outputs even if cryosuction is not included in modeling. However, at 1 m depth,





error can increase to 10% ~ 30% by simulation without cryosuction representation.

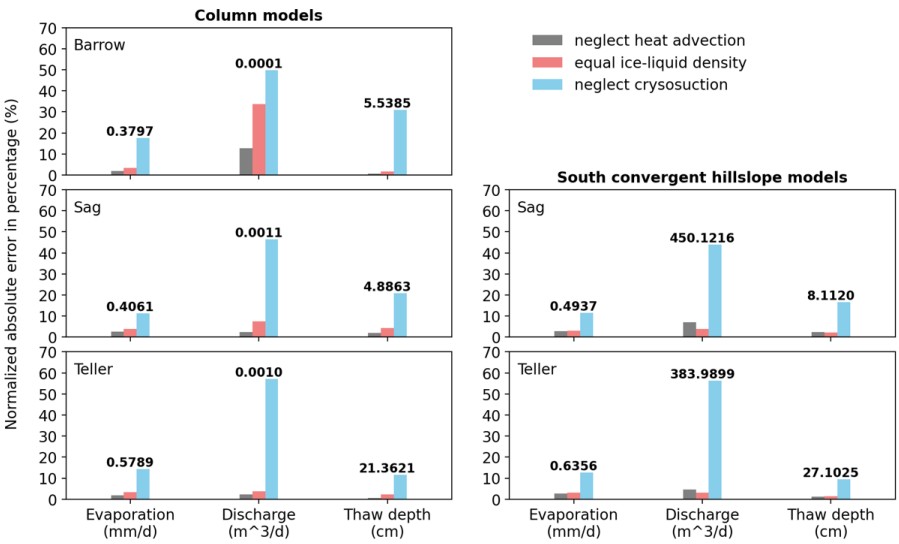


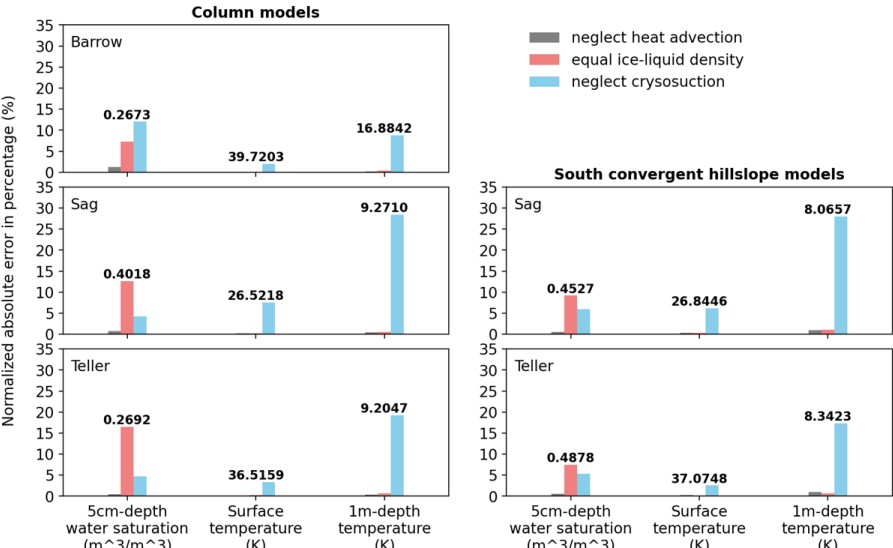

**Figure 13 Normalized average absolute error of variables over the period of 2011-2020, compared among**
**three physical assumptions at column and hillslope scales. Variables: (a) evaporation, discharge, and thaw**
**depth, numbers in figures are the average values of the corresponding variables; (b) water saturation, and**
**temperature, numbers in figures are the average annual fluctuation range of each variable.**
**5 Conclusion**



Simplification of Arctic process representation is an essential consideration when developing
process-rich models for thermal permafrost hydrology. There are three subsurface processes that
are commonly described in a simplified approach for many Arctic tundra models: (i) ice is
prescribed the same density as liquid water; (ii) the effect of soil cryosuction is neglected; (iii)
advective heat transport is neglected. Here we investigated the influence of these simplified
representations on modeling field-scale permafrost hydrology.
To do this, we conducted an ensemble of simulations using the Advanced Terrestrial Simulator
(ATS v1.2) to evaluate the impact of the above three process simplifications on field-scale
predictions. The ensemble of simulations consisted of 62 numerical experiments considering
various conditions, including different climate conditions and soil properties at three sites of
Alaska, and different model scale conceptualizations. For evaluation, we compared integrated
variables (evaporation, discharge), averaged thaw depth, and pointwise variables (temperature,
total water saturation), among different models to access the deviation of applying a simplified
modeling assumption. The main conclusions of this study are summarized as follows:
1) Excluding soil cryosuction in permafrost models can cause significant bias in most

hydrological variables. Especially, according to this study, the average deviation in

evaporation, discharge, and thaw depth may reach $10\% \sim 20\%$, $50\% \sim 60\%$, and $10\% \sim$

$30\%$, respectively, relative to the corresponding annual average values. The prediction

error for discharge may grow if rainfall rates increase. In the case of pointwise variables,

the error in temperature increases from a small amount at the surface up to $10\% \sim 30\%$ at

1 m beneath surface. The prediction of subsurface temperature and water saturation is

especially affected when considering hillslope scale models. Therefore, soil cryosuction

should be included when modeling permafrost change.

2) Assuming equal ice-liquid density will not result in especially large deviations when

predicting most of the hydrological variables, particularly at hillslope scales. It primarily

affects the prediction of soil water saturation profile and can cause $5\% \sim 15\%$ error relative

to the annual saturation fluctuation range. This difference may have consequences for the

carbon cycle with regards to the production of methane versus carbon dioxide. Assigning

liquid water density for ice may reduce computational time to a small extent in ATS,

dependent on simulating conditions and spatial and temporal scales.

3) For a general Arctic tundra system, the prediction error in most variables after neglecting





advective heat transport is less than 5%, or even much lower. In the case of ATS, the

simulation time cost for hillslope-scale models can decrease by 40% to 80% under

conditions in this study. Ignoring heat advection in the absence of local, flow-focusing

mechanisms, such as thermo-erosion gullies, seems a reasonable decision.

Through the comparison of permafrost hydrological outputs obtained from ensemble model setup
targeted at field scale, we confirm the importance and necessity of including soil cryosuction effect
in predicting permafrost changes, and valid application of equal ice-liquid density and neglecting
advective heat transport for a general Arctic system. The latter two may also ease computational
cost dependent upon simulation conditions. We expect that this study can contribute to the
development of permafrost hydrology models, as well as better selection of physical process
representations for modelers.
**Code availability**
Advanced Terrestrial Simulator (ATS) is an open-source code for solving ecosystem-based,
integrated, distributed hydrology, and available at https://github.com/amanzi/ats. Simulations were
conducted using version 1.2 (Coon et al., 2021).
**Data availability**
Data sources of wind speed are cited in the text. The raw forcing data acquired from Daymet, the
processed forcing data used for simulation, and simulation output data are available through
https://github.com/gaobhub/data_for_paper_model_comparison.
**Author contributions**
Bo Gao did some revision of the code to add options for process representations, designed
numerical experiments and setup models, did data analysis and interpretation, drafted and revised
the article. Ethan T. Coon implemented the code in which the study was done, conceptualized the
study, helped debug the runs, and helped draft and revise the article.
**Competing interests**
The authors declare that they have no conflict of interest.



**Acknowledgement**
Both authors are supported by the U.S. Department of Energy, Office of Science, Biological and
Environmental Research program under the InteRFACE project. This research used resources of
the Compute and Data Environment for Science (CADES) at the Oak Ridge National Laboratory,
which is supported by the Office of Science of the U.S. Department of Energy under Contract No.
DE-AC05-00OR22725.

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
