# Peer review of "Evaluating Simplifications of Subsurface Process"

_The Cryosphere, 2021_

## Referee Comment (RC2)

[revised manuscript text omitted]

*[margin note: these are not cast as "processes" here, They are here cast as "process simplifications"]*

To do this, we conducted an ensemble of simulations using the Advanced Terrestrial Simulator (ATS v1.2) to evaluate the impact of the above three process simplifications on field-scale predictions. The ensemble of simulations consisted of 62 numerical experiments considering various conditions, including different climate conditions and soil properties at three sites of

Alaska, and different model scale conceptualizations. For evaluation, we compared integrated variables (evaporation, discharge), averaged thaw depth, and pointwise variables (temperature, total water saturation), among different models to access the deviation of applying a simplified modeling assumption. The main conclusions of this study are summarized as follows:

1) Excluding soil cryosuction in permafrost models can cause significant bias in most hydrological variables.  *[margin: In particular, under the assumed conditions of]* this study, the average deviation in evaporation, discharge, and thaw depth may reach 10% ~ 20%, 50% ~ 60%, and 10% ~

30%, respectively, relative to the corresponding annual average values. The prediction error for discharge may grow if rainfall rates increase. In the case of pointwise variables, the error in temperature increases from a small amount at the surface up to 10% ~ 30% at

1 m beneath surface. The prediction of subsurface temperature and water saturation is especially affected when considering hillslope scale models. Therefore, soil cryosuction should be included when modeling permafrost change.

2) Assuming equal ice-liquid density will not result in especially large deviations when predicting most of the hydrological variables, particularly at hillslope scales. It primarily affects the prediction of *[the]* soil water saturation profile and can cause 5% ~ 15% error relative to the annual saturation fluctuation range. This difference may have consequences for the carbon cycle with regards to the production of methane versus carbon dioxide. Assigning liquid water density for ice may reduce computational time to a small extent in ATS, dependent on simulating conditions and spatial and temporal scales.

3) For a general Arctic tundra system, the prediction error in most variables after neglecting

[Figure]

*All results were focussed on the errors in making these simplifications, more discussion would help in identifying why some were more sensitive than others, i.e. need to provide more insight into the role of the physical processes. Also - no spatial results shown of 3D model?*

[revised manuscript text omitted]

---

## Author Comment (AC1)

Dear Reviewer,

Thank you very much for taking the time to review our work. We greatly appreciate your thoughtful comments that help improve the manuscript. Our responses to the comments and how we will revise the manuscript based on these comments are listed as follows.

**RC #1.**
*The key question is, if the content of the paper is of sufficient general interest and is really giving new insight. The title of the paper suggests that the results are generally relevant for "field-scale permafrost hydrology models". However, they will to some extent be influenced by the concrete modelling approach, discretization scheme, linear solver (the authors mention that the AMG-preconditioner they use is not well suited for advective transport). Thus it is more a kind of sensibility analysis of the results produced by their code in different scenarios. The authors tend to not carefully distinguish between small differences in the model results produced by their code and a low relevance of the process in reality (or at least in modelling reality).*

**AC #1.**
We completely agree that this is crucial for establishing the importance of the paper, have tried to do so in this submission, and will continue to improve this in the subsequent revision.

The premise of this paper is that, by starting from general equations of mass and energy transport and simplifying those equations, we can use a model to understand the relative importance of given transport mechanisms in describing the physics of permafrost hydrology. Under that premise, this paper is relevant for both users of models and for non-modelers looking to understand the importance of physical processes. Models are unique in their ability to change physical assumptions and perform counterfactual experiments (e.g., what if there was no advective heat transport?). We believe that the key results, such as that "cryosuction is an important process for explaining the soil moisture in Arctic soils at hillslope scales", are important results of interest to more than just modelers. We will improve upon this message in the paper revision.

However, considering the approach that we used to present these results in this work, we will clarify this general interest by changing the title to: "Evaluating Simplifications of Subsurface Process Representations for Field-scale Permafrost Hydrology with ATS (v1.2)". We recognize and acknowledge that there are limitations to this generality due to the spatial-temporal scales and model configuration, and various numerical codes.

Firstly, for the scales and model configurations, simplified pseudo-3D hillslopes without focusing mechanisms representing headwaters, hilly terrains, and columns representing vertically-dominated, flat landscapes with limited heterogeneity are frequently used in the hydrologic community for understanding watershed function and field-scale observations. Therefore, both of the configurations were chosen in this paper. They are different with those smaller scales (e.g., polygonal ice wedge) or configurations (e.g., taliks, thermos-erosion gullies, etc.) which are usually applied for studying localized features. We have tried to make this clear in the Introduction by specifically citing literature where advective heat transport has been demonstrated to be crucial. In our revision, we will make this clearer by emphasizing this point and citing literature working

in these scales.

Secondly, from the perspective of numerical codes, this work is most relevant to users and developers of permafrost hydrology models. In addition to ATS, there have been several codes with proven capability to address permafrost or frozen soil relevant problems, as we showed in the Introduction. Though the specific representation of a physical process and its implementation in code could be different, or some processes are not considered in some codes, the principal physics (e.g., mass and energy transport) among these codes are similar, and these results are especially relevant to that class of models.

However, we disagree that the discretization scheme or numerical methods are relevant to the process comparison results. If discretization error is shown to be smaller than the physical "error" of excluding a process, then surely discretization scheme can be eliminated (see more on this below). While differences in linear solvers will alter the computational efficiency of the computation, it will not alter the solution of the governing partial differential equation, given that each solution is made to a given tolerance. We do agree that these are crucial considerations in considering the computational performance of a code, and agree that the performance numbers are ATS-specific and not relevant to other models or codes. While these performance numbers are of relevance to a much smaller audience, we thought they were worth including as a sidenote.

**RC #2.**
*What is also missing is an analysis of the discretization error associated with the different grids and the time step used.*

**AC #2.**
We agree that, in order to deem a process "important," it is necessary to demonstrate that the differences due to process representation are larger than the errors introduced via discretization, and agree that this is an important step in justifying the results. To confirm this, a grid convergence study was conducted, and the discretization error was compared with the "error" caused by omitting a given process. This comparison was done for the column mesh and the southern-aspect divergent hillslope mesh under the Sag River conditions. Thaw depth, as a significant permafrost concept, was used as the evaluation quantity for the comparison.

For the column model (50 m deep), we generated five meshes. From the coarsest to the finest, the numbers of cells are 20, 39, 78, 156, 312. We used 78 cells in the paper. Column models with these different numbers of cells were conducted with full physics representations. For the 78-cells column model, we also conducted simulations with simplified representation for each physical process, i.e., equal liquid-ice density, neglecting advective heat transport, and neglecting cryosuction effect, which has been discussed in the paper. The column model with the finest mesh was assumed to output the most accurate results. All other column models with full physics representations were compared to the finest model. The 10-year averaged absolute error in thaw depth was calculated and shown in Figure S1. Figure S1 illustrates the relation between error and numbers of cells in double logarithmic coordinates. Black points are discretization error compared the finest column model, which are almost on the same line, demonstrating the expected first order convergence rate (due to first-order upwinding methods being used in both the advective term and

the relative permeability). The blue, red, and green points show the errors caused by each physics simplification, respectively. Clearly the error due to process omission is comparable to the discretization error in the first two cases, but not in the cryosuction case. This supports the conclusion that cryosuction is crucial. It also demonstrates that our measured differences in advective heat transport and equal liquid and ice densities are, at best, upper bounds on the true differences – they may in fact be smaller.

[Figure]

**S1. Average error in thaw depth caused by numbers of cells and physical process representations for column models under site Sag's condition**

Additionally, we also performed the same grid convergence study for a hillslope model. The south-facing divergent hillslope mesh was selected to maximize the dynamic range of the system. We generated five meshes with different numbers of cells, from the coarsest to the finest, which are 1050, 1950, 3900, 7800, 15600, where 3900 cells were used for hillslope meshes in the paper. The five hillslope models were conducted with full physics representations, and we also have the 3900-cell hillslope model with simplified physics representations as discussed in the paper. The results obtained using the finest mesh was considered the most accurate and the other four models with coarser meshes were compared with it. The black points in Figure S2 are the 10-year averaged absolute errors in thaw depth for the four hillslope models with different numbers of meshes. They are almost on the same line demonstrating the spatial resolution has little impact on solutions. Similar results to the column model are shown, though the error associated with neglecting advective heat transport and liquid-ice density have grown slightly relative to the discretization error, hinting (but not conclusively proving) that these may be closer to true measures of the error than upper bounds on the error.

[Figure]

**S2. Average error in thaw depth caused by number of cells and physical process representations for southern aspect divergent hillslope models under site Sag's condition.**

The above contents will be added to the Supplement.

**RC #3.**
*Line 191: The soil-freezing characteristic curve is usually used as a material property of a certain soil. Thus I find this term here rather confusing.*

**AC #3.**
Different constitutive relationships describe the partitioning of water into liquid and ice in different ways. We show two different ways of determining the liquid, water, and gas saturations as a function of pressure and temperature – Equations (3) (Painter and Karra, 2014) and Equaiton (4) (McKenzie et al., 2007). In fact, there are many other models and many previous studies have presented various SFCC relationships, empirically or related soil-water characteristic curve (e.g., Ren et al., 2017; Stuurop et al., 2021).

We agree with the reviewer that the parameters involved in these various SFCC relationships are properties of the soil. Different soils may show different SFCC shape even under the same model, and different models will result in different physical behavior of the system. Crucially for this paper, Equation (3) admits cryosuction (see Painter & Karra, 2014) while Equation (4) does not admit cryosuction. We will clarify this in the revised manuscript.

**RC #4.**

*Figure 1 (Precipitation and air temperature of site Barrow, Sag, and Teller from year 2011 to 2020): Too much information is packaged into too small figures here. It is very hard to see for example the rain precipitation at sag, because it is in the background of the other sites.*

**AC #4.**

Figure 1 shows daily, as well as annually averaged precipitation and air temperature. The purpose of presenting this figure is to conceptually show that the three fields we chose have different climate conditions, and we agree that this was too much information to accomplish this goal. Daily precipitation/temperature data will be removed, and only annually averaged data will be shown in Figure 1 as follows.

[Figure]

**Figure 1 Precipitation and air temperature of site Barrow, Sag, and Teller from year 2011 to 2020**

**RC #5.**

*Table 2 (Soil properties of three soil layers of all sites used in this paper): The van Genuchten-Mualem model can produce unphysical results for n values much smaller than 2, which is true for all parameter sets here.*

**AC #5.**

All van Genuchten parameters of soils used in this paper are from previous permafrost studies, which have been calibrated or measured. Atchley et al. (2015), Jafarov et al. (2018), and O'Connor et al. (2020) presents these parameters in site Barrow, Teller, and Sag, respectively. We agree that the Mualem model can introduce nonphysical results for relative permeability for n values much smaller than 2; in each case, this "infinite slope" is avoided by introducing a spline function matching the derivative and value of the relative permeability at a saturation of 0.95. This approach is used frequently in codes solving Richards equation using the Mualem model. Care must be taken to ensure that this spline is monotonically increasing as it approaches saturated conditions; this is true for all cases included here.

**RC #6.**

*Line 333-344: How was this column initialization transferred to the hillslope? Does this not produce an instable initial value for the hillslope?*

**AC #6.**

The first step initialization is to obtain a steady-state frozen soil column, the water table of which is almost just below the column top surface. The second step initialization is to use this frozen soil column to do fully coupled run, which solves integrated surface-subsurface, mass/energy balance system with smoothed forcing data to obtain annual cyclic equilibrium state. The final temperature and pressure profile of the column is then assigned to each column of the hillslope mesh because the vertical discretization of the hillslope mesh is the same as the column. In this way, we have the initial state of a hillslope, and use it to solve the fully coupled system. We agree that this is unstable initially on a hillslope, but, in our experience, this is rectified in the first year or two of the simulation. The results of the first year were ignored in case of any unstable solutions.

**RC #7.**

*Line 374-37: Might this averaging of local data not smooth the effect of neglecting processes? If you have only a local effect at one or two points, this could be greatly reduced by the averaging.*

**AC #7.**

We agree with the reviewer that influence could be greater at one location than another. So, we designed the hillslope mesh with a changing slope from the hillslope top to the toe, and chose five different surface locations along the changing slope to record data there instead of averaging thaw depth or surface temperature across the whole surface domain. We did compare the simulated results at different points (at both surface and subsurface). The differences caused by process representations among the five different observation points are similar. This paper presents the comparison using the averaged results at those selected locations, as we believe this is more representative of the information desired by a reader than showing the maximum or "worst case" error, which might exaggerate a difference that is spatially (or temporally) limited. This is consistent with the use of these models in the literature, where typically integrated or averaged parameters, such as ET, runoff, active layer thickness, etc. are of more interest.

**RC #8.**

*Figure 4 and 5: The figures are again rather small.*

**AC #8.**

All figures in the text will be modified to a higher resolution by changing figure size, font size of texts, and colors of lines.

**RC #9.**

*Figure 6: The concept of a "decrease percentage" is rather hard to understand (especially if it gets negative). Would it not be easier to understand, if you use the relative runtime? Which than would be either smaller than one (thus faster) or larger?*

**AC #9.**

We apologize for the poor choice of words; we were in fact showing relative runtime change (in

percentage). The captions and axis labels of Figure 6 and Figure 12 will be revised to "relative runtime change".

Take Figure 6 as an example:
The old caption of Figure 6: Decreased percentage of simulation time under the assumption of equal ice-liquid density compared to the real ice density representation for all hillslope scale simulations.
Revised caption of Figure 6: The relative runtime change in percentage due to the assumption of equal ice-liquid density compared to that with the real ice density representation for all hillslope scale models.

**RC #10.**
*Section 4.2: There is no information about the effect of neglecting cryosuction on the runtime. However, isn't that the main point of the paper (how much precision do you sacrifice for which acceleration?)*

**AC #10.**
The principle of selecting a process representation is, under the premise of correct simulation output with acceptable errors, to improve computing efficiency. Neglecting cryosuction results in large deviations in hydrological estimations, especially for thaw depth which is a critical feature of permafrost region. The deviations may get larger in wetter conditions. So cryosuction effect should be included in permafrost models even if it may require additional computational cost. Besides, based on the hillslope scale simulations in this work, including cryosuction effect may or may not slow down simulation depending on soil properties and conditions. The cases that can speed up simulation when excluding cryosuction just decrease the running time by 10%~30%, much smaller than that we see in the thermal advection section. We agree that this should be included for completeness. However, considering the computational cost is analyzed just based on ATS code and primarily useful for ATS users, and this is a much smaller audience, we will move the discussions of the computational cost to the Appendix. The manuscript is probably clearer with their removal from the main text.

**RC #11.**
*Figure 13: I am not sure, how much this figure really helps in understanding. you need to read the text very carefully to only understand, what is represented here (not talking about what it means).*

**AC #11.**
We agree that we present a lot of data in this paper, and we have tried to simplify it as much as possible while still making clear that the results are complete. We were torn between the conflicting goals of 1, being concise and clear and 2, demonstrating that we were not omitting metrics or results that would have supported the inclusion of advective transport and density variation. It is much easier to show that a process must be represented than to convince readers that a given process probably doesn't need to be included.

This figure was intended to be a "big picture" view, comparing relative differences across processes and metrics. We hope that the reader sees this figure and immediately grasps that the blue lines (representing cryosuction) are bigger, in almost all cases, across metrics. We will revise the labels and caption of the figure to make it clearer as follows:

[Figure]

**Figure 13 Percentage errors for each metric caused by physics simplifications at column and hillslope scales. The percentage error refers to the averaged error of a metric over the period of 2011-2020 normalized by a certain reference value obtained from full-physics model. Metrics include (a) evaporation, discharge, and thaw depth; (b) water saturation, and temperatures. Numbers in figures are the corresponding reference values for each metric: (a) 10-year average obtained from full-physics model; (b) 10-year averaged annual fluctuation range obtained from full-physics model.**

**RC #12.**

*Conclusions: The results found here do not have to be representative for all "permafrost models" Thus conclusions like "Excluding soil cryosuction in permafrost models can..." or "Assuming equal ice-liquid density will not result..." are a bit ambitious or even dangerous.*

**AC #12.**

Conclusions will be revised in the text to add "under the assumed conditions in this paper" to avoid too broad and general conclusions. We agree that it is important to state the limitations of these conclusions, and we have tried to clarify this in the text. As stated in the initial response, however, we do think that these conclusions are properties of the PDEs being solved, and not of the specific implementation of the code. Besides, this work uses ATS (v1.2) as a tool to examine whether including or excluding one physical process in permafrost models will make a significant impact on hydrological outputs for a large-scale Arctic system without apparent influence caused by localized groundwater flow features, such as taliks, thermal-erosion gullies etc., that mentioned in the Introduction. It cannot represent all codes that also address permafrost problems, but it can provide a reference for code development.

**RC #13.**

*Line 494: "Factitiously" is a very rare word. how about "artificially"?*

**AC #13.**

This will be revised in the text.

**Reference**

Atchley, A. L., Painter, S. L., Harp, D. R., Coon, E. T., Wilson, C. J., Liljedahl, A. K., and Romanovsky, V. E.: Using field observations to inform thermal hydrology models of permafrost dynamics with ATS (v0.83), Geosci. Model Dev., 8, 2701–2722, https://doi.org/10.5194/gmd-8-2701-2015, 2015.

Jafarov, E. E., Coon, E. T., Harp, D. R., Wilson, C. J., Painter, S. L., Atchley, A. L., and Romanovsky, V. E.: Modeling the role of preferential snow accumulation in through talik development and hillslope groundwater flow in a transitional permafrost landscape, Environ. Res. Lett., 13, 105006, https://doi.org/10.1088/1748-9326/aadd30, 2018.

McKenzie, J. M., Voss, C. I., and Siegel, D. I.: Groundwater flow with energy transport and water–ice phase change: numerical simulations, benchmarks, and application to freezing in peat bogs, Adv. Water Resour., 30, 966–983, https://doi.org/10.1016/j.advwatres.2006.08.008, 2007.

O'Connor, M. T., Cardenas, M. B., Ferencz, S. B., Wu, Y., Neilson, B. T., Chen, J., and Kling, G. W.: Empirical Models for Predicting Water and Heat Flow Properties of Permafrost Soils, Geophys. Res. Lett., 47, e2020GL087646, https://doi.org/10.1029/2020GL087646, 2020.

Painter, S. L. and Karra, S.: Constitutive Model for Unfrozen Water Content in Subfreezing Unsaturated Soils, Vadose Zone J., 13, vzj2013.04.0071, https://doi.org/10.2136/vzj2013.04.0071, 2014.

Ren, J., Vanapalli, S., and Han, Z.: Soil freezing process and different expressions for the soil-freezing characteristic curve, Sci. Cold Arid Reg., 9, 221–228, https://doi.org/10.3724/SP.J.1226.2017.00221, 2017.

Stuurop, J. C., van der Zee, S. E. A. T. M., Voss, C. I., and French, H. K.: Simulating water and heat transport with freezing and cryosuction in unsaturated soil: Comparing an empirical, semi-empirical and physically-based approach, Adv. Water Resour., 149, 103846, https://doi.org/10.1016/j.advwatres.2021.103846, 2021.

---

## Author Comment (AC2)

Dear Reviewer,

Thank you very much for taking the time to review our work. We greatly appreciate your thoughtful comments that help improve the manuscript. Our responses to the comments and how we will revise the manuscript based on these comments are listed as follows.

**RC #1.**

*All results are shown as time series and error plots. More insight is needed into the actual physical processes and system behavior, not just on 'dry' figures or plots showing errors. i.e. to answer WHY these processes are or are not important under these conditions.*

**AC #1.**

We agree that there is room for more discussion of the processes and why results change these processes representations. We will add the following analyses in the text, and figures were added to the Supplement.

(1) Why models with cryosuction predict deeper thaw depth?

Essentially, cryosuction increases soil suction to attract more deep liquid water moving towards the frozen front during soil freezing. Thus, the real active layer formed due to the existence of cryosuction should be thicker than the cases in which cryosuction is assumed unimportant.

(2) Why Barrow site is more sensitive to the cryosuction process when estimating thaw depth and water content (Figure 8 of the manuscript)?

This is determined by both soil properties and climate conditions. The soil at Barrow has larger suction and is able to hold more water (Figure 2 of the manuscript), providing the possibility for cryosuction to make larger contributions. The principal difference between cryosuction and non-cryosuction representations is presented when temperature is below the freezing point (see Eq.(3) and Eq.(4) of the manuscript). Compared to Sag and Teller, Barrow has lower annual average temperature (Figure 1 of the manuscript), making the effect of cryosuction more pronounced.

(3) Why Sag and Teller sites are more sensitive to the cryosuction process when estimating temperature (Figure 9 of the manuscript)?

This is associated with the larger water present at these two sites. Soil freezes from ground surface downward and from the bottom of active layer upward during freezing, forming a liquid zone in between where the temperature approximates freezing point due to phase change (The following Figure S1(a) shows an example of the column model under Sag's condition at the 300[th] day of one year). Thus, this liquid zone isolates the upper permafrost from the soil surface temperature variations due to the weakened conductive heat transport the along soil depth. Besides, the released latent heat in this liquid zone may retard soil freezing, which also tends to reduce the thermal conduction. However, cryosuction process can speed up freezing and promote the attenuation of the liquid zone, and thus decrease the impact of the liquid zone (Figure S1(b) shows the ice

saturation at the same time, i.e., 300[th] of one year, when the soil column still has large area no frozen). Hence, the influence of cryosuction is more significant with more soil water.

[Figure]

**Figure S1. (a) Ice saturation and area where temperature is between 273.15K and 273.25K within the top 1 m depth of a column model under Sag's condition at DOY = 300, with cryosuction process in simulation (DOY is day of year); (b) Ice saturation within the top 1 m depth of a column model under Sag's condition at DOY = 300, without cryosuction process in simulation.**

(4) Why advective heat transport is not significant in simulations under the assumed conditions of this paper?

Under the assumption of large-scale Arctic systems ignoring influence by localized groundwater flow features (e.g., ponds, gullies, etc.), the liquid water flux determines the advective heat transport in the subsurface. However, the flow velocity on average is quite low within the shallow active layer with limited thickness (see an example in Figure S2).

[Figure]

**Figure S2. Vertical velocity distribution and thaw depth within the top 1m depth of a column model under Sag's condition at DOY = 208 and 240.**

Figure S3 compares the absolute value of conductive and advective heat flux, at the 208[th] and 240[th] days, separately. The advective heat flux only shows a relatively larger value at top cells because of water flow inside of the active layer. The relatively larger advective heat flux is on the same order of magnitude with the smallest conductive heat flux (see Figure S3(a)) or even less than the smallest conductive heat flux (see Figure S3(b)).

[Figure]

**Figure S3. Absolute value of conductive and advective heat flux within the top 1m depth of a column model under Sag's condition at DOY = 208 and 240.**

**RC #2.**

*I did not find the comparison of computational efficiency very relevant. The authors seem to suggest if the computational cost of including advective heat transport is high, then it can be neglected. Computational cost should have little or no bearing on whether or not to include a process - if a process is important & relevant, it needs to be included, regardless of the computational cost.*

**AC #2.**

We agree that the process distinction is by far the more interesting result, and we included the computational cost only as a sidenote, as it is of interest only to ATS users. We definitely did not intend to suggest that computational cost has any say in whether a process can be neglected or not. We agree with the reviewer that if a process is important and relevant, computational cost should not be a deciding factor. Considering the computational cost is analyzed just based on ATS code and primarily useful for ATS users, and this is a much smaller audience, we will move the discussions of the computational cost to the Appendix. The manuscript is probably clearer with their removal from the main text.

**RC #3.**

*I found the results and conclusions were cast too strongly as being definitive in the general context. These results are specific for the conditions assumed (geometry, flow system, etc.).*

**AC #3.**

We agree that the results are specific for the conditions considered, and tried to stress this in both the Introduction and Conclusions by noting clear exceptions in other geometries, such as thermo-erosion gullies, etc. (see Lines 118-128, Lines 608-609). We agree that it important to state the limitations of this study up front, however, and will add text at the beginning of the conclusions clarifying that "Here we investigated the influence of these simplified representations on modeling field-scale permafrost hydrology in set of simplified geometries commonly used in the permafrost

hydrology literature with the Advanced Terrestrial Simulator (ATS v1.2). We note that these conclusions are specific to conditions similar to these geometries, and should not be applied in cases where focusing flow mechanisms may dominate." Additionally, in the Conclusions, we will also add "under the assumed conditions".

**RC #4.**
*The paper refers a few times to 'a general Arctic system' (Line 27) or to '... a normal Arctic system' (Line 490) .... These should be replaced by, ex., 'a conceptual system'... or 'in these specific simplified cases'. There is no such thing as a 'general' or 'normal' Arctic system.*

**AC #4.**
The "general/normal Arctic system" here does not mean "generalized" or "all" Arctic system, but refer to a large-scale Arctic system without apparent influence caused by localized groundwater flow features, such as taliks, thermal-erosion gullies etc., that mentioned in the Introduction. For a small area with these localized features, advective heat transport may play an important role. This paper does not focus on these localized features, but on a large-scale Arctic system where the influence of these features can be neglected. This will be clarified in the text.

**RC #5.**
*Line 111: The Nixon (1975) paper is much too old to use for justifying this statement that 'it is commonly recognized that heat conduction predominates ...'.*

**AC #5.**
This sentence was intended to lead to the introduction of the cases where advective heat transport plays an important role. It will be deleted in the text.

**RC #6.**
*Line 156, 297: needs to be corrected to advection-dispersion (or advection-conduction). ('diffusion' is almost always used only in the context of mass transport).*

**AC #6.**
This is clearly a difference of fields. Advection-diffusion is commonly used in the applied mathematics literature to describe the partial differential equation solved, though we are willing to consider that conduction may be the more commonly used term in the engineering community, and is typically more specifically used when referring to heat transport. The term "advection-diffusion" in the original manuscript (Line 156) will be revised to "advection-conduction". Section 2.3 focuses on advective heat transport and discusses the effect of including advective heat transport or not in permafrost models. Section 2.3 will remain the original heading "advective heat transport".

**RC #7.**
*Table 6 (A summary of NNSEs of variables obtained through column model comparison): four significant digits is excessive here.*

**AC #7.**
Four digits were used to avoid "NNSE = 1.000". This will be revised to three digits in the text.

**RC #8.**
*Line 184: $s_n$ (saturation of n-phase) is usually capitalized.*

**AC #8.**
Lowercase "*s*" for saturation is used to differ from the function $S_*$ for Van Genuchten model.

---

## Author Response (AR1)

**Response to reviewer comments 1**

Dear Reviewer,

Thank you very much for taking the time to review our work. We greatly appreciate your thoughtful comments that help improve the manuscript. Below we attempt to answer the points of critics from the reviewer more or less point by point.

**RC #1.1**

*The key question is, if the content of the paper is of sufficient general interest and is really giving new insight. The title of the paper suggests that the results are generally relevant for "field-scale permafrost hydrology models". However, they will to some extent be influenced by the concrete modelling approach, discretization scheme, linear solver (the authors mention that the AMG-preconditioner they use is not well suited for advective transport). Thus, it is more a kind of sensibility analysis of the results produced by their code in different scenarios. The authors tend to not carefully distinguish between small differences in the model results produced by their code and a low relevance of the process in reality (or at least in modelling reality).*

**AC #1.1**

The main concern of Reviewer #1 was that the title may be too broad relative to the paper content, and the reviewer expressed concerns that the conclusions that drew based on the model sensitivity analysis may be limited to one model or one code. We completely agree that this is crucial for establishing the importance of the paper.

First, the premise of this paper is that, by starting from general equations of mass and energy transport and simplifying those equations, we can use a model to understand the relative importance of given transport mechanisms in describing the physics of permafrost hydrology. Under that premise, this paper is relevant for both users of models and for non-modelers looking to understand the importance of physical processes. Models are unique in their ability to change physical assumptions and perform counterfactual experiments (e.g., what if there was no advective heat transport?). We believe that the key results, such as that "cryosuction is an important process for explaining the soil moisture in Arctic soils at hillslope scales", are important results of interest

to more than just modelers. **We have clarified this point in the revised manuscript, see:**

- The end of the second paragraph of the Introduction
- The end of the last paragraph of the Introduction
- The first paragraph of the Conclusion

Second, specifically speaking of the model configurations we setup in this study, hillslope models can represent headwaters and hilly terrains; column models represent vertically dominated, flat landscapes with limited heterogeneity. These two kinds of model configurations are frequently used in the hydrologic community for understanding watershed function and field-scale observations. Therefore, both of the configurations were chosen in this paper. They are different with those smaller scales (e.g., polygonal ice wedge) or configurations (e.g., taliks, thermos-erosion gullies, etc.) which are usually applied for studying localized features. We have tried to make this clear in the Introduction by specifically citing literature where advective heat transport has been demonstrated to be crucial. **In the revised manuscript, we have further clarified the conclusions made from the advective heat transport analysis by emphasizing the scale limitation, see:**

- The abstract
- The end of the 5$^{th}$ paragraph of the Introduction
- The first paragraph of section 4.3 (Advective heat transport)
- The last two paragraphs of the Conclusion

Third, from the perspective of numerical codes, this work is most relevant to users and developers of permafrost hydrology models. In addition to ATS, there have been several codes with proven capability to address permafrost or frozen soil relevant problems, as we showed in the Introduction. Though the specific representation of a physical process and its implementation in code could be different, or some processes are not considered in some codes, the principal physics (e.g., mass and energy transport) among these codes are similar, and these results are especially relevant to that class of models. **This point is included to the first paragraph of the Conclusion.**

We have clarified the limitations of this work by making our introduction, main contents, and conclusions clearer in the paper. We are confident that we have made a clear case that this work

and its conclusions are of interest not just to ATS and its users, but to permafrost hydrologists in general, as the conclusions speak to the physical processes studied, and not just the model or code. Therefore, we believe that the existing title is more appropriate than including the code name, which would detract from that argument by putting focus on the code itself. While we agree that there are limitations to this title, largely due to model scales, we believe these have sufficiently been addressed in other edits to the text.

Finally, for the other concern the reviewer mentioned "*small differences in the model results produced by their code and a low relevance of the process in reality*", however, we disagree that the discretization scheme or numerical methods are relevant to the process comparison results. If discretization error is shown to be smaller than the physical "error" of excluding a process, then surely discretization scheme can be eliminated (**see more details in the response AC #2**). While differences in linear solvers will alter the computational efficiency of the computation, it will not alter the solution of the governing partial differential equation, given that each solution is made to a given tolerance. We do agree that these are crucial considerations in considering the computational performance of a code, and agree that the performance numbers are ATS-specific and not relevant to other models or codes. While these performance numbers are of relevance to a much smaller audience, we thought they were worth including as a sidenote.

**RC #1.2**
*What is also missing is an analysis of the discretization error associated with the different grids and the time step used.*

**AC #1.2**
We agree that, in order to deem a process "important," it is necessary to demonstrate that the differences due to process representation are larger than the errors introduced via discretization, and agree that this is an important step in justifying the results. To confirm this, a grid convergence study was conducted, and the discretization error was compared with the "error" caused by omitting a given process. This comparison was done for the column mesh and the southern-aspect divergent hillslope mesh under the Sag River conditions. Thaw depth, as a significant permafrost

concept, was used as the evaluation quantity for the comparison. **All of the following grid convergence analysis has been added to the Supplement.**

For the column model (50 m deep), we generated five meshes. From the coarsest to the finest, the numbers of cells are 20, 39, 78, 156, 312. We used 78 cells in the paper. Column models with these different numbers of cells were conducted with full physics representations. For the 78-cells column model, we also conducted simulations with simplified representation for each physical process, i.e., equal liquid-ice density, neglecting advective heat transport, and neglecting cryosuction effect, which has been discussed in the paper. The column model with the finest mesh was assumed to output the most accurate results. All other column models with full physics representations were compared to the finest model. The 10-year averaged absolute error in thaw depth was calculated and shown in Figure S1. Figure S1 illustrates the relation between error and numbers of cells in double logarithmic coordinates. Black points are discretization error compared the finest column model, which are almost on the same line, demonstrating the expected first order convergence rate (due to first-order upwinding methods being used in both the advective term and the relative permeability). The blue, red, and green points show the errors caused by each physics simplification, respectively. Clearly the error due to process omission is comparable to the discretization error in the first two cases, but not in the cryosuction case. This supports the conclusion that cryosuction is crucial. It also demonstrates that our measured differences in advective heat transport and equal liquid and ice densities are, at best, upper bounds on the true differences – they may in fact be smaller.

[Figure]

**S1. Average error in thaw depth caused by numbers of cells and physical process representations for column models under site Sag's condition**

Additionally, we also performed the same grid convergence study for a hillslope model. The south-facing divergent hillslope mesh was selected to maximize the dynamic range of the system. We generated five meshes with different numbers of cells, from the coarsest to the finest, which are 1050, 1950, 3900, 7800, 15600, where 3900 cells were used for hillslope meshes in the paper. The five hillslope models were conducted with full physics representations, and we also have the 3900-cell hillslope model with simplified physics representations as discussed in the paper. The results obtained using the finest mesh was considered the most accurate and the other four models with coarser meshes were compared with it. The black points in Figure S2 are the 10-year averaged absolute errors in thaw depth for the four hillslope models with different numbers of meshes. They are almost on the same line demonstrating the spatial resolution has little impact on solutions. Similar results to the column model are shown, though the error associated with neglecting advective heat transport and liquid-ice density have grown slightly relative to the discretization error, hinting (but not conclusively proving) that these may be closer to true measures of the error than upper bounds on the error.

[Figure]

**S2. Average error in thaw depth caused by number of cells and physical process representations for southern aspect divergent hillslope models under site Sag's condition.**

**RC #1.3**

*Line 191 (original manuscript): The soil-freezing characteristic curve is usually used as a material property of a certain soil. Thus, I find this term here rather confusing.*

**AC #1.3**

Different constitutive relationships describe the partitioning of water into liquid and ice in different ways. We show two different ways of determining the liquid, water, and gas saturations as a function of pressure and temperature – Equations (3) (Painter and Karra, 2014) and Equation (4) (McKenzie et al., 2007). In fact, there are many other models and many previous studies have presented various soil-freezing characteristic curve (SFCC) relationships, empirically or related soil-water characteristic curve (e.g., Ren et al., 2017; Stuurop et al., 2021). We agree with the reviewer that the parameters involved in these various SFCC relationships are properties of the soil. Different soils may show different SFCC shape even under the same model, and different models will result in different physical behavior of the system. Crucially for this paper, Equation (3) admits cryosuction (see Painter & Karra, 2014) while Equation (4) does not admit cryosuction. **We have clarified the concepts and different representations of cryosuction in the section 2.2 (Cryosuction).**

**RC #1.4**

*Figure 1 (Precipitation and air temperature of site Barrow, Sag, and Teller from year 2011 to 2020): Too much information is packaged into too small figures here. It is very hard to see for example the rain precipitation at sag, because it is in the background of the other sites.*

**AC #1.4**

Figure 1 shows daily, as well as annually averaged precipitation and air temperature. The purpose of presenting this figure is to conceptually show that the three fields we chose have different climate conditions, and we agree that this was too much information to accomplish this goal. **Figure 1 has been modified in the revised manuscript**.

**RC #1.5**

*Table 2 (Soil properties of three soil layers of all sites used in this paper): The van Genuchten-Mualem model can produce unphysical results for n values much smaller than 2, which is true for all parameter sets here.*

**AC #1.5**

All van Genuchten parameters of soils used in this paper are from previous permafrost studies, which have been calibrated or measured. Atchley et al. (2015), Jafarov et al. (2018), and O'Connor et al. (2020) presents these parameters in site Barrow, Teller, and Sag, respectively. We agree that the Mualem model can introduce nonphysical results for relative permeability for n values much smaller than 2; in each case, this "infinite slope" is avoided by introducing a spline function matching the derivative and value of the relative permeability at a saturation of 0.95. This approach is used frequently in codes solving Richards equation using the Mualem model. Care must be taken to ensure that this spline is monotonically increasing as it approaches saturated conditions; this is true for all cases included here.

**RC #1.6**

*Line 333-344 (original manuscript): How was this column initialization transferred to the hillslope? Does this not produce an instable initial value for the hillslope?*

**AC #1.6**

The first step initialization is to obtain a steady-state frozen soil column, the water table of which is almost just below the column top surface. The second step initialization is to use this frozen soil column to do fully coupled run, which solves integrated surface-subsurface, mass/energy balance system with smoothed forcing data to obtain annual cyclic equilibrium state. The final temperature and pressure profile of the column is then assigned to each column of the hillslope mesh because the vertical discretization of the hillslope mesh is the same as the column. In this way, we have the initial state of a hillslope, and use it to solve the fully coupled system. We agree that this is unstable initially on a hillslope, but, in our experience, this is rectified in the first year or two of the simulation. The results of the first year were ignored in case of any unstable solutions. **The initialization process has been clarified in the second paragraph of section 3.3 (Model setup).**

**RC #1.7**

*Line 374-37 (original manuscript): Might this averaging of local data not smooth the effect of neglecting processes? If you have only a local effect at one or two points, this could be greatly reduced by the averaging.*

**AC #1.7**

We agree with the reviewer that influence could be greater at one location than another. So, we designed the hillslope mesh with a changing slope from the hillslope top to the toe, and chose five different surface locations along the changing slope to record data there instead of averaging thaw depth or surface temperature across the whole surface domain. We did compare the simulated results at different points (at both surface and subsurface). The differences caused by process representations among the five different observation points are similar. This paper presents the comparison using the averaged results at those selected locations, as we believe this is more representative of the information desired by a reader than showing the maximum or "worst case" error, which might exaggerate a difference that is spatially (or temporally) limited. This is consistent with the use of these models in the literature, where typically integrated or averaged parameters, such as ET, runoff, active layer thickness, etc. are of more interest.

**RC #1.8**

*Figure 4 and 5: The figures are again rather small.*

**AC #1.8**

All figures in the text have been revised to a higher resolution.

**RC #1.9**

*Figure 6 (original manuscript): The concept of a "decrease percentage" is rather hard to understand (especially if it gets negative). Would it not be easier to understand, if you use the relative runtime? Which than would be either smaller than one (thus faster) or larger?*

**AC #1.9**

We apologize for the poor choice of words; we were in fact showing relative runtime change (in percentage). **All runtime figures and analysis has been moved to the Appendix. The captions and axis labels of these figures have been revised (see the Appendix).**

**RC #1.10**

*Section 4.2: There is no information about the effect of neglecting cryosuction on the runtime. However, isn't that the main point of the paper (how much precision do you sacrifice for which acceleration?)*

**AC #1.10**

The principle of selecting a process representation is, under the premise of correct simulation output with acceptable errors, to improve computing efficiency. Neglecting cryosuction results in large deviations in hydrological estimations, especially for thaw depth which is a critical feature of permafrost region. The deviations may get larger in wetter conditions. So cryosuction effect should be included in permafrost models even if it may require additional computational cost. Besides, based on the hillslope scale simulations in this work, including cryosuction effect may or may not slow down simulation depending on soil properties and conditions. The cases that can speed up simulation when excluding cryosuction just decrease the running time by 10%~30%, much smaller than that we see in the thermal advection section. We agree that this should be included for completeness. However, considering the computational cost is analyzed just based on ATS code and primarily useful for ATS users, and this is a much smaller audience, we moved the discussions of the computational cost to the Appendix, **see the Appendix for all runtime figures and analysis**. The manuscript is probably clearer with their removal from the main text.

**RC #1.11**

*Figure 13 (original manuscript): I am not sure, how much this figure really helps in understanding. you need to read the text very carefully to only understand, what is represented here (not talking about what it means).*

**AC #1.11**

We agree that we present a lot of data in this paper, and we have tried to simplify it as much as possible while still making clear that the results are complete. We were torn between the conflicting goals of 1, being concise and clear and 2, demonstrating that we were not omitting metrics or results that would have supported the inclusion of advective transport and density variation. It is much easier to show that a process must be represented than to convince readers

that a given process probably doesn't need to be included. This figure was intended to be a "big picture" view, comparing relative differences across processes and metrics. We hope that the reader sees this figure and immediately grasps that the blue lines (representing cryosuction) are bigger, in almost all cases, across metrics. The labels and caption of the figure has been revised in the text, **see Figure 11 in the revised manuscript**.

**RC #1.12**

*Conclusions: The results found here do not have to be representative for all "permafrost models" Thus conclusions like "Excluding soil cryosuction in permafrost models can..." or "Assuming equal ice-liquid density will not result..." are a bit ambitious or even dangerous.*

**AC #1.12**

We agree that it is important to state the limitations of these conclusions, and we have tried to clarify this in the text. As stated in the initial response, however, we do think that these conclusions are properties of the PDEs being solved, and not of the specific implementation of the code. Besides, this work uses ATS (v1.2) as a tool to examine whether including or excluding one physical process in permafrost models will make a significant impact on hydrological outputs for a large-scale Arctic system without apparent influence caused by localized groundwater flow features, such as taliks, thermal-erosion gullies etc., that mentioned in the Introduction. It cannot represent all codes that also address permafrost problems, but it can provide a reference for code development. **Limitations of this work has been added to the Conclusion, see:**

- The first paragraph of the Conclusion
- The end of the 2$^{nd}$ paragraph of the Conclusion
- Results summaries from the 3$^{rd}$ through the 6$^{th}$ paragraph of the Conclusion
- The last paragraph of the Conclusion.

**RC #1.13**

*Line 494 (original manuscript): "Factitiously" is a very rare word. how about "artificially"?*

**AC #1.13**

This has been revised in the text, **see the first paragraph of section 3.3 (Model setup), and the first paragraph of section 4.3 (Advective heat transport).**

**Reference**

Atchley, A. L., Painter, S. L., Harp, D. R., Coon, E. T., Wilson, C. J., Liljedahl, A. K., and Romanovsky, V. E.: Using field observations to inform thermal hydrology models of permafrost dynamics with ATS (v0.83), Geosci. Model Dev., 8, 2701–2722, https://doi.org/10.5194/gmd-8-2701-2015, 2015.

Jafarov, E. E., Coon, E. T., Harp, D. R., Wilson, C. J., Painter, S. L., Atchley, A. L., and Romanovsky, V. E.: Modeling the role of preferential snow accumulation in through talik development and hillslope groundwater flow in a transitional permafrost landscape, Environ. Res. Lett., 13, 105006, https://doi.org/10.1088/1748-9326/aadd30, 2018.

McKenzie, J. M., Voss, C. I., and Siegel, D. I.: Groundwater flow with energy transport and water–ice phase change: numerical simulations, benchmarks, and application to freezing in peat bogs, Adv. Water Resour., 30, 966–983, https://doi.org/10.1016/j.advwatres.2006.08.008, 2007.

O'Connor, M. T., Cardenas, M. B., Ferencz, S. B., Wu, Y., Neilson, B. T., Chen, J., and Kling, G. W.: Empirical Models for Predicting Water and Heat Flow Properties of Permafrost Soils, Geophys. Res. Lett., 47, e2020GL087646, https://doi.org/10.1029/2020GL087646, 2020.

Painter, S. L. and Karra, S.: Constitutive Model for Unfrozen Water Content in Subfreezing Unsaturated Soils, Vadose Zone J., 13, vzj2013.04.0071, https://doi.org/10.2136/vzj2013.04.0071, 2014.

Ren, J., Vanapalli, S., and Han, Z.: Soil freezing process and different expressions for the soil-freezing characteristic curve, Sci. Cold Arid Reg., 9, 221–228, https://doi.org/10.3724/SP.J.1226.2017.00221, 2017.

Stuurop, J. C., van der Zee, S. E. A. T. M., Voss, C. I., and French, H. K.: Simulating water and heat transport with freezing and cryosuction in unsaturated soil: Comparing an empirical, semi-empirical and physically-based approach, Adv. Water Resour., 149, 103846, https://doi.org/10.1016/j.advwatres.2021.103846, 2021.

**Response to reviewer comments 2**

Dear Reviewer,

Thank you very much for taking the time to review our work. We greatly appreciate your thoughtful comments that help improve the manuscript. Below we attempt to answer the points of critics from the reviewer more or less point by point.

**RC #2.1**

*All results are shown as time series and error plots. More insight is needed into the actual physical processes and system behavior, not just on 'dry' figures or plots showing errors. i.e. to answer WHY these processes are or are not important under these conditions.*

**AC #2.1**

We agree that there is room for more discussion of the processes and why results change these processes representations. In the revised manuscript, we have added the following analyses. Relevant figures were added to the Supplement.

- **The second paragraph of section 4.2 (Cryosuction) was rewritten and extended to discuss the following two questions:**
  (1) Why models with cryosuction predict deeper thaw depth?
  (2) Why Barrow site is more sensitive to the cryosuction process when estimating thaw depth and water content?

- **The third paragraph of section 4.2 (Cryosuction) was extended to discuss the following question**:
  (1) Why Sag and Teller sites are more sensitive to the cryosuction process when estimating temperature?

- **The first paragraph of section 4.3 (Advective heat transport) was extended to discuss the following question**:
  (1) Why advective heat transport is not significant in simulations under the assumed conditions of this paper?

**RC #2.2**

*I did not find the comparison of computational efficiency very relevant. The authors seem to suggest if the computational cost of including advective heat transport is high, then it can be neglected. Computational cost should have little or no bearing on whether or not to include a process - if a process is important & relevant, it needs to be included, regardless of the computational cost.*

**AC #2.2**

We agree that the process distinction is by far the more interesting result, and we included the computational cost only as a sidenote, as it is of interest only to ATS users. We definitely did not intend to suggest that computational cost has any say in whether a process can be neglected or not. We agree with the reviewer that if a process is important and relevant, computational cost should not be a deciding factor. Considering the computational cost is analyzed just based on ATS code and primarily useful for ATS users, and this is a much smaller audience, **we have moved the discussions of the computational cost to the Appendix.** The manuscript is probably clearer with their removal from the main text.

**RC #2.3**

*I found the results and conclusions were cast too strongly as being definitive in the general context. These results are specific for the conditions assumed (geometry, flow system, etc.).*

**AC #2.3**

We agree that the results are specific for the conditions considered. In the original manuscript (first submission), we tried to stress this in both the Introduction and Conclusions by noting clear exceptions in other geometries, such as thermo-erosion gullies, etc. (see Lines 118-128, Lines 608-609 in the original manuscript). We agree that it important to state the limitations of this study up front. **Limitations have been added to the main contents and the Conclusion, see:**

- In the analysis of the influence of the advective heat transport, it has been clarified through the manuscript that situations in which localized groundwater flow plays an important role were excluded, see the abstract, the end of the 5$^{th}$ paragraph of the Introduction, the first

paragraph of section 4.3 (Advective heat transport), and the last two paragraphs of the Conclusion.

- The precondition of the conclusions drew from the sensitive analyses of this study were added to the end of the second paragraph of the Conclusion.
- Results summaries from the 3$^{rd}$ through the 6th paragraph of the Conclusion.

**RC #2.4**

*The paper refers a few times to 'a general Arctic system' (Line 27) or to '... a normal Arctic system' (Line 490) .... These should be replaced by, ex., 'a conceptual system'... or 'in these specific simplified cases'. There is no such thing as a 'general' or 'normal' Arctic system.*

**AC #2.4**

The "general/normal Arctic system" here does not mean "generalized" or "all" Arctic system, but refer to a large-scale Arctic system without apparent influence caused by localized groundwater flow features, such as taliks, thermal-erosion gullies etc., that mentioned in the Introduction. For a small area with these localized features, advective heat transport may play an important role. This paper does not focus on these localized features, but on a large-scale Arctic system where the influence of these features can be neglected.

In the revised manuscript, all phrases like "generalized", "normal" Arctic system have been replaced by "system where localized groundwater flow features can be neglected" or similar expressions, **see the abstract, the end of the 5$^{th}$ paragraph of the Introduction, the first paragraph of section 4.3 (Advective heat transport), and the last two paragraphs of the Conclusion.**

**RC #2.5**

*Line 111 (original manuscript): The Nixon (1975) paper is much too old to use for justifying this statement that 'it is commonly recognized that heat conduction predominates ...'.*

**AC #2.5**

This sentence was intended to lead to the introduction of the cases where advective heat transport plays an important role. **It has been deleted in the text, see the 5$^{th}$ paragraph of the Introduction.**

**RC #2.6**

*Line 156, 297 (original manuscript): needs to be corrected to advection-dispersion (or advection-conduction). ('diffusion' is almost always used only in the context of mass transport).*

**AC #2.6**

This is clearly a difference of fields. Advection-diffusion is commonly used in the applied mathematics literature to describe the partial differential equation solved, though we are willing to consider that conduction may be the more commonly used term in the engineering community, and is typically more specifically used when referring to heat transport. **The term "advection-diffusion" in the original manuscript has been revised to "advection-conduction", see the second paragraph of section 2 (Theory).** Section 2.3 focuses on advective heat transport and discusses the effect of including advective heat transport or not in permafrost models. So, **the original heading "advective heat transport" of section 2.3 should be more appropriate.**

**RC #2.7**

*Table 6 (A summary of NNSEs of variables obtained through column model comparison): four significant digits is excessive here.*

**AC #2.7**

Four digits were used to avoid "NNSE = 1.000". **The table has been revised to three digits in the text, see Table 6 in the revised manuscript.**

**RC #2.8**

*Line 184 (original manuscript): $s_n$ (saturation of n-phase) is usually capitalized.*

**AC #2.8**

Lowercase "$s$" for saturation is used to differ from the function $S_*$ for van Genuchten model.